# Hallucination is Inevitable:
# An Innate Limitation of Large Language Models

### Abstract

Hallucination has been widely recognized to be a significant drawback for large language models (LLMs). There have been many works that attempt to reduce the extent of hallucination. These efforts have mostly been empirical so far, which cannot answer the fundamental question whether it can be completely eliminated. In this paper, we formalize the problem and show that it is impossible to eliminate hallucination in LLMs. Specifically, we define a formal world where hallucination is defined as inconsistencies between a computable LLM and a computable ground truth function. By employing results from learning theory, we show that LLMs cannot learn all the computable functions and will therefore inevitably hallucinate if used as general problem solvers. Since the formal world is a part of the real world which is much more complicated, hallucinations are also inevitable for real world LLMs. Furthermore, for real world LLMs constrained by provable time complexity, we describe the hallucination-prone tasks and empirically validate our claims. Finally, using the formal world framework, we discuss the possible mechanisms and efficacies of existing hallucination mitigators as well as the practical implications on the safe deployment of LLMs.

## 1 Introduction

The emergence of large language models (LLMs) has marked a significant milestone in the field of artificial intelligence, particularly in natural language processing. These models, with their vast knowledge bases and ability to generate coherent and contextually relevant text, have greatly impacted research, industry, and society. However, one of the critical challenges they face is the problem of "*hallucination,*" where the models generate plausible but factually incorrect or nonsensical information. This issue has brought increasing concerns about safety and ethics as LLMs are being applied widely, resulting in a growing body of literature trying to classify, understand, and mitigate it.

Prior works have identified multiple possible sources of hallucination in LLMs from the data collection to the training and inference aspects. For example, in the survey paper (Ji et al., 2023), the authors attribute hallucination in natural language generation to heuristic data collection, innate divergence, imperfect representation learning, erroneous decoding, exposure bias, and parametric knowledge bias. A plethora of methods have been proposed to mitigate hallucination. For example, factual-centred metrics (Goodrich et al., 2019; Guerreiro et al., 2023; Mishra et al., 2021; Shuster et al., 2021) and benchmarks (Li et al., 2023b; Lin et al., 2022; Vu et al., 2023) have been proposed to measure and reduce hallucination on specific datasets. Retrieval-based methods reinforce LLM by knowledge graphs or databases to help correct factual errors (Shuster et al., 2021; Zhao et al., 2023). Prompting the models to reason (Wei et al., 2022b) and verify (Dhuliawala et al., 2023) their answers has also been shown to reduce hallucination.

Up to now, research on LLM hallucination remains largely empirical. Useful as they are, empirical studies cannot answer the fundamental question: *can hallucination be completely eliminated*? The answer to this question is fundamental as it indicates a possible upper limit of LLMs' abilities. However, since it is impossible to empirically enumerate and test every possible input, formal discussion on this question is impossible without a clear definition and formal analysis of hallucination.

In the real world, formally defining hallucination, a factual or logical error of LLM, turns out to be extremely difficult. This is because a formal definition of semantics in the real world is still an open problem (van Deemter, 2024; Speaks, 2021). Hence in this work, we rigorously define a formal world of computable functions, wherein precise discussions on hallucination is feasible. In this world, hallucination occurs whenever an LLM fails to exactly reproduce the output of a computable function. Under this definition, we present a fundamental result that *hallucination is inevitable for any computable LLM, regardless of model architecture, learning algorithms, prompting techniques, or training data.* Since this formal world is a part of the real world, the result also applies to LLMs in the real world. To the best of our knowledge, this is the first work that formally defines hallucination and discusses its inevitability. A parallel work by Kalai and Vempala (Kalai & Vempala, 2024) provided a statistical lower bound on the rate of hallucination for calibrated LLMs. It shows a significant fact that pretraining LLMs for predictive accuracy leads to hallucination even with perfect training data. The results in this paper is comparably more general as they are applicable to all computable LLMs.

The contributions of this paper are: (1) we formally define and discuss hallucination for LLMs and, by employing results in learning theory (Bārzdiņš & Freivald, 1972; Bazhenov et al., 2020; Gold, 1967; Jain et al., 1999), we show that hallucination is inevitable for all computable LLMs; (2) we discuss practical implications of theoretical results on the hallucination mitigators and deployment of LLMs in the real world; (3) we show examples of hallucination-prone problems and validate the claims by empirical study.

## 2 Definitions

In this section, we detail our definitions of LLMs, hallucination, and training and deployment of LLMs under the hallucination context. Readers are referred to Appendix A for a table of notation.

### 2.1 Large Language Model

We start with the definitions for our alphabet and strings:

**Definition 1** (Alphabet and Strings)**.** Let $\mathbb{N}$ be the set of natural numbers. An alphabet $\mathcal{A}$ is a finite set of $N$ tokens $\mathcal{A} = \{a_0, a_1, \ldots, a_{N-1}\}$. A string is a sequence $w_{0:n-1} = w_0 w_1 \ldots w_{n-1}$ obtained by concatenating tokens for $n$ times, where $i, n, N \in \mathbb{N}, w_i \in \mathcal{A}$. Let $\mathcal{S}$ be a computable set[1] of all the finite-length strings of alphabet $\mathcal{A}$ and $(s_0, s_1, \ldots)$ be an one-to-one enumeration of all the elements in $\mathcal{S}$.

For real-world LLMs, tokens are usually unique combinations of alphanumerical characters. An LLM is a probabilistic model of a string that conditions the output at time $t$ based on all the tokens that come before it in the string. The most common usage of LLM is to complete a partial string $w_{0:q-1}$, or a prompt, into a complete string $w_{0:n-1}$, where $n > q$, by maximizing the likelihood of the latter in a token-by-token manner. Readers are referred to Appendix B for implementation details of LLMs.

To facilitate more general discussions, we now ignore all the implementation details of an LLM and only focus on its input and output behavior after training. Though the current LLMs are resource constrained, in this paper we consider trained LLMs to be total computable functions (usually, as functions from $\mathcal{S}$ to $\mathcal{S}$).

An LLM is trained to adapt to training samples before being deployed (a general training procedure is detailed in Procedure 1 in Section 2.3). This means an LLM has different states depending on the training samples it has seen in a training process[2]. We use $h^{[i]}$ to denote the state of LLM $h$ at the $i^{th}$ stage of training (for example, after the LLM is trained on $i$ samples $\{(s_0, f(s_0)), (s_1, f(s_1)), \ldots, (s_{i-1}, f(s_{i-1}))\}$). We assume that the training process as well as all the states of LLMs after training are uniformly computable (that is, given an LLM and the training samples, we can effectively obtain the program of an LLM state after it is trained on certain part of the training samples at some stage).

---

[1]A computable set (Turing, 1937) is a set whose membership is decidable by a computable function.
[2]In practice, most LLMs are discussed in an offline setting, where it is trained on a certain set of finite data and then deployed without further training. This is a special case of the setting in this paper, where we consider hallucination for different states of a given LLM based on different sets of training samples.

All real-world LLMs have some properties, for example, they all complete their computation in polynomial time. In our discussion below, we will also consider computably enumerable sets of LLMs where all the member LLMs have a particular property $P$. Following the definition, a set of $P$-property LLMs is a proper subset of all LLMs. This classification is useful for our further discussion about LLMs' limitations, where we will view different LLMs with different properties as different subsets of total computable functions.

Different from general total computable functions, LLMs can be categorized by the desirability of its outputs along a spectrum. On the "nonsensical" side is a clueless token predictor which produces meaningless outputs given input strings $s$. On the "ideal" side, a hallucination-free function produces sensible and factual outputs for any well-formed input strings. In between are the real-world LLMs: their outputs are comprehensible most of the time; however, they occasionally "hallucinate" and generate nonfactual statements. This spectrum and the relation between concepts in this section is shown in Fig. 1.

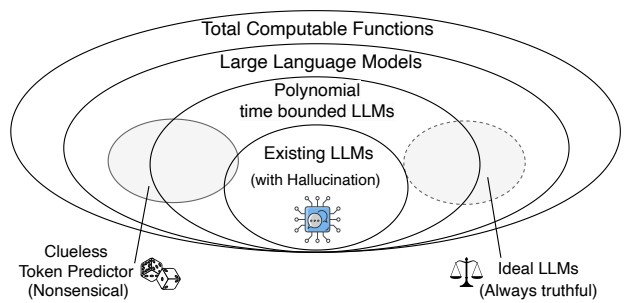

Figure 1: Relations between LLMs, $P$-property LLM given some $P$ (e.g., $P$ is polynomial time bounded), and computable functions. The set of ideal LLMs is not attainable, as will be shown.

### 2.2 A Formal World and Hallucination

Hallucination is in essence an erroneous output produced by an LLM. Without getting tangled in the difficult problem of formalizing "correctness" in our real world, we define hallucination in a formal world where $f$ is the ideal function that produces correct output (e.g., completion) $f(s)$ for any input string (or prompt, question etc.) $s \in \mathcal{S}$.

**Definition 2** (Formal World of $f$)**.** A formal world of ground truth function $f$ is a set $\mathcal{G}_f = \{(s, f(s)) \mid s \in \mathcal{S}\}$, where $f(s)$ is the only correct output of input string $s$, for all $s \in \mathcal{S}$.

The training samples $\mathcal{T}$ is defined as a set of input-output pairs we get from the formal world of $f$:

**Definition 3** (Training Samples $\mathcal{T}$)**.** Training samples $\mathcal{T}$ is a set $\{(s_0, y_0), (s_1, y_1), \ldots, (s_i, y_i), \ldots \mid y_i = f(s_i), s_i \in \mathcal{S}, i \in \mathbb{N}\}$.

Set $\mathcal{T}$ is a generalized corpus of what $f$ outputs given input strings. For example, if $f$ answers "true" for factual inputs and "false" otherwise, then $\mathcal{T}$ could look like { ("A shark is a mammal.", "false"), ("Earth orbits around the Sun.", "true"), ... }. On the other hand, if $f$ is a function that completes or answers input string $s$, then $\mathcal{T}$ could look like { ("Is shark a fish or mammal?", "Fish."), ("What is the sum of binary numbers 10 and 11?", "101."), ... }.

With the ground truth $f$ introduced, it is straightforward to define hallucination in the formal world as when all the states of an LLM fails to fully reproduce the output of ground truth function $f$. Formally:

**Definition 4** (Hallucination)**.** An LLM $h$ is hallucinating with respect to a ground truth function $f$, if $\forall i \in \mathbb{N}, \exists s \in \mathcal{S}$ such that $h^{[i]}(s) \neq f(s)$.

### 2.3 Training an LLM

In the formal world, using our definition of hallucination in Definition 4, the core question of whether hallucination can be eliminated can be translated to:

**Question 1** (The Fundamental Question)**.** Can an LLM $h$ be trained following a fixed procedure, such that for any ground-truth function $f$, $\exists i \in \mathbb{N}, \forall s \in \mathcal{S}, h^{[i]}(s) = f(s)$?

We are not interested in the training and deployment details like model initialization, selection of optimizers, learning rates, objective functions, stopping criteria, inference hyperparameters (e.g., temperature,) and so on, and wish our discussion to be independent of these factors. Therefore, we lump together all the factors

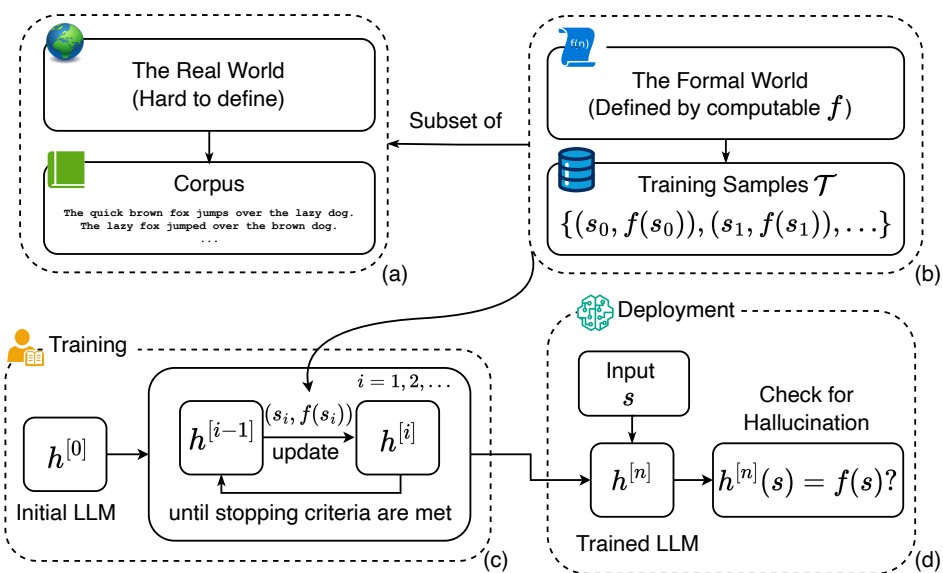

Figure 2: Illustration of relations between all concepts defined in Section 2. (a) shows the real-world corpus, which is a superset of (b) the formal world of ground truth function $f$ and training samples $\mathcal{T}$. (c) shows the procedure that trains LLM $h$ as defined in Procedure 1, which takes training samples and updates $h$ until the stopping criteria are met. Finally in (d), the LLM at a state $n$ is deployed and produces output given an unseen string $s$. Hallucination is defined by comparing LLM's answer $h^{[n]}(s)$ with ground truth value $f(s)$.

and call it a "training and deploying" procedure. In this procedure, an LLM $h$ is iteratively updated by training samples. We check the LLM for hallucination during deployment, after the stopping criteria are met and the training procedure stops. Formally:

---

**Procedure 1** (Training and deploying an LLM)**.**

- Input: A *stream*[a] of training samples $\mathcal{T} = ((s_0, f(s_0)), (s_1, f(s_1)), \ldots)$.

- Output: $h^{[i]}$, an LLM in a trained state indexed by $i$, where $i \in \mathbb{N}$.

- Procedure:

    1. Let $i = 0$. Let $h^{[0]}$ be the LLM in the initial state (e.g., with randomly initialized parameters).
    2. Training and Validation Iteration:
        (a) If stopping criteria are met (LLM is ready): end iteration, goto step 3.
        (b) Retrieve a training sample $(s_i, f(s_i))$ from $\mathcal{T}$.
        (c) Update LLM $h^{[i]}$ to $h^{[i+1]}$ according to training samples $\{(s_j, f(s_j)) \mid j \leq i\}$.
        (d) Let $i \leftarrow i + 1$, go to step 2a.

    3. Deployment: Let $h^{[i]}$ be the final trained state. End the procedure.

    ---
    [a]Assuming no limitations on their numbers.

---

There are a few notes about this procedure:

- In step 2a, traditional stopping criteria include a preset number of iterations or stagnating loss functions. The procedure above does not assume any criterion, meaning the training can take arbitrarily many finite samples and arbitrarily long finite time.

- As a result of this procedure, $h^{[i]}$ is a state of LLM $h$ in which it is trained on samples $\{(s_j, f(s_j)) \mid j < i\}$. In particular, $h^{[0]}$ is an LLM state in which $h$ is initialized but never trained.

- The trained LLM $h^{[i]}$ can be expected to: (1) answer $f(s) = ?$, or (2) given "$(s,$" complete it with "$f(s))$", or (3) given "is it true that $f(s) = t$ ?" answer "yes" or "no". For ease of presentation and without loss of generality, we assume case (1), where LLM uses $h^{[i]}(s)$ to answer the question.

An illustration showing the relations between all the definitions in this section is shown in Fig. 2. It is noteworthy that LLMs trained by Procedure 1 are far more powerful and flexible than their counterparts in the real world. Therefore, *if hallucination is inevitable for our LLMs in the relatively simple formal world, then hallucination is inevitable for LLMs in the more complicated real world.*

## 3 Hallucination is Inevitable for LLMs

In this section, we show that hallucination is inevitable in the formal world by suitably applying results in learning theory literature. Specifically, we will use the diagonalization argument in our proof. The diagonalization argument was originally used to prove that some infinite sets (e.g., the set of real numbers) are larger than others (e.g., the set of natural numbers) by Cantor (Cantor, 1890/91; Ewald, 2005). The high-level idea is to show that any enumeration of an uncountable set is not exhaustive by constructing a counter example that is in the set but not in the enumeration (Wikipedia contributors, 2023). The argument gets its name because such counter example is constructed by flipping the diagonal entries, i.e., the $i^{th}$ component of the $i^{th}$ element, in the table showing the enumeration.

This section is organized in a restricted-to-general manner, where we first discuss less complex LLMs resembling real-world ones and then generalize to any computable LLMs. *First*, we show that all states of all the LLMs in any computably enumerable sets will hallucinate on some inputs (Theorem 1). *Then*, we show that all states of all the LLMs in a computably enumerable set will hallucinate on infinite many inputs (Theorem 2). *Finally*, we show that hallucination is inevitable for all computable LLMs (Theorem 3). At the end, we will answer the fundamental Question 1.

### 3.1 Computably Enumerable LLMs will Hallucinate

As all the currently proposed LLMs are polynomial time bounded, these LLMs (considered as functions) are contained in a computably enumerable set of LLMs. To prove that this set of LLMs must hallucinate, below we prove a stronger theorem that *any* computably enumerable set of LLMs (including this set) will inevitably hallucinate, using the classic diagonalization argument.

**Theorem 1.** For all computably enumerable sets of LLMs $\{h_0, h_1, \dots\}$, there exists a computable ground truth function $f$, such that all $h_i^{[j]}, i, j \in \mathbb{N}$, will hallucinate.

Table 1: Illustration of diagonalization for the proof of Theorem 1. Each row represents a particular LLM state. Each column represents a string $s_i$ with which the LLM is tested. Blue cells are the outputs used to construct $f$ in Eq. (1)

| LLMs | Test Samples | | | | |
|---|---|---|---|---|---|
| | $s_0$ | $s_1$ | $s_2$ | $s_3$ | $\cdots$ |
| $\hat{h}_0$ | $\boldsymbol{\hat{h}_0(s_0)}$ | $\hat{h}_0(s_1)$ | $\hat{h}_0(s_2)$ | $\hat{h}_0(s_3)$ | $\cdots$ |
| $\hat{h}_1$ | $\hat{h}_1(s_0)$ | $\boldsymbol{\hat{h}_1(s_1)}$ | $\hat{h}_1(s_2)$ | $\hat{h}_1(s_3)$ | $\cdots$ |
| $\hat{h}_2$ | $\hat{h}_2(s_0)$ | $\hat{h}_2(s_1)$ | $\boldsymbol{\hat{h}_2(s_2)}$ | $\hat{h}_2(s_3)$ | $\cdots$ |
| $\hat{h}_3$ | $\hat{h}_3(s_0)$ | $\hat{h}_3(s_1)$ | $\hat{h}_3(s_2)$ | $\boldsymbol{\hat{h}_3(s_3)}$ | $\cdots$ |
| $\vdots$ | $\vdots$ | $\vdots$ | $\vdots$ | $\vdots$ | $\ddots$ |
| $f$ | $\boldsymbol{\Delta(\hat{h}_0(s_0))}$ | $\boldsymbol{\Delta(\hat{h}_1(s_1))}$ | $\boldsymbol{\Delta(\hat{h}_2(s_2))}$ | $\boldsymbol{\Delta(\hat{h}_3(s_3))}$ | $\cdots$ |

*Proof.* Consider any computably enumerable set of LLMs $\{h_0, h_1, \dots\}$. An LLM $h_i$ can be trained on no training sample, one training sample, two training samples, and so on, resulting in its different

states $\{h_i^{[0]}, h_i^{[1]}, \ldots, h_i^{[j]}, \ldots\}, i, j \in \mathbb{N}$. To facilitate diagonalization argument, we use the Cantor pairing function (Cantor, 1878) to re-enumerate all these LLM states as $\{\hat{h}_0, \hat{h}_1, \ldots, \hat{h}_k, \ldots\}$, where $h_i^{[j]}$ is re-enumerated as $\hat{h}_k$ with $k = (i + j)(i + j + 1)/2 + j, \forall i, j \in \mathbb{N}$. In order to demonstrate the contradiction, let us assume that at least one of the LLM states is hallucination-free (if there is none, the theorem is trivially proved). Feeding a string $s$ to LLMs in all the enumerated states, we get their outputs as $\{\hat{h}_0(s), \hat{h}_1(s), \ldots\}$. Repeating this for all $s \in \mathcal{S}$, we get a table as in Table 1, which contains all the outputs of all the states of all the LLMs in the set being considered.

Now we define the ground truth function $f$ so that $f(s_i)$ contradicts the elements along the diagonal of the table (see blue cells in Table 1):

$$f(s_i) = \Delta\big(\hat{h}_i(s_i)\big), \forall i \in \mathbb{N}, \tag{1}$$

where $\Delta$ is a computable function that returns a different string from its input. For example, we could define $\Delta\big(s_k\big) = s_{k+1}$, which returns the next string of $s_k$ in $\mathcal{S}$. Therefore $\Delta\big(\hat{h}_i(s_i)\big) \neq \hat{h}_i(s_i)$. By the above construction and according to Definition 4, $\hat{h}_0$ hallucinates w.r.t. $f$ as $f(s_0) \neq \hat{h}_0(s_0)$ and so for $\hat{h}_1$ as $f(s_1) \neq \hat{h}_1(s_1)$. Repeating this reasoning for all $s_i \in \mathcal{S}$, we see that all states of all the LLMs in the given set will hallucinate w.r.t. $f$ because $f(s_i) \neq \hat{h}_i(s_i)$ for all $i \in \mathbb{N}$. This contradicts with our assumption and thus proves the theorem. $\qquad\square$

**LLMs will Hallucinate on Infinitely Many Questions**   One might argue that Theorem 1 only shows that all states of all the LLMs in any computably enumerable set will hallucinate on one but not all inputs. Therefore, hallucination in the formal world could be negligible. However, we can show the following theorem:

**Theorem 2.** For all computably enumerable sets of LLMs $\{h_0, h_1, \ldots\}$, there exists a computable ground truth function $f$, such that all $h_i^{[j]}, i, j \in \mathbb{N}$, hallucinate on infinitely many inputs.

*Proof.* Consider the following computably enumerable set of LLMs: $\{h_0, h_1, \ldots\}$. Similar to the proof of Theorem 1, we enumerate all the states of these LLMs as $\{\hat{h}_k | k \in \mathbb{N}\}$ using the Cantor pairing function. Feeding a string $s$ to these LLMs, we get their answers as $\{\hat{h}_0(s), \hat{h}_1(s), \ldots\}$.

Define a computable function $f$ as follows:

$$f(s_i) = \Delta\big(\{\hat{h}_j(s_i) \mid j \leq i\}\big), \forall i \in \mathbb{N},$$

where $\Delta\big(\{\hat{h}_j(s_i) \mid j \leq i\}\big)$ returns a string that is different from all the strings in set $\{\hat{h}_j(s_i) \mid j \leq i\}$.

By the above construction, $\forall i \geq j, f(s_i) \neq \hat{h}_j(s_i)$ because the value of $f(s_i)$ is explicitly defined as a string different from $\hat{h}_j(s_i), i \geq j$ (see Table 2). Therefore, $\hat{h}_0$ will hallucinate on $s_0, s_1, \ldots$; and $\hat{h}_1$ will hallucinate on $s_1, s_2, \ldots$. In general, LLM $\hat{h}_k$ will hallucinate on all input strings after $s_{k-1}$ in the one-to-one enumeration $(s_0, s_1, \ldots)$ of $\mathcal{S}$. $\qquad\square$

Table 2: Illustration of proof for Theorem 2. Each row represents a particular LLM. Each column represents a string $s_i$ where the LLM is tested on. A value in the $j^{th}$ column of $i^{th}$ row represents $\hat{h}_i(s_j)$. Blue cells are the outputs that we use to construct $f$, on which the LLM in the corresponding row will hallucinate.

| LLMs | Test Samples | | | | |
|---|---|---|---|---|---|
| | $s_0$ | $s_1$ | $s_2$ | $s_3$ | $\cdots$ |
| $\hat{h}_0$ | $\hat{h}_0(s_0)$ | $\hat{h}_0(s_1)$ | $\hat{h}_0(s_2)$ | $\hat{h}_0(s_3)$ | $\cdots$ |
| $\hat{h}_1$ | $\hat{h}_1(s_0)$ | $\hat{h}_1(s_1)$ | $\hat{h}_1(s_2)$ | $\hat{h}_1(s_3)$ | $\cdots$ |
| $\hat{h}_2$ | $\hat{h}_2(s_0)$ | $\hat{h}_2(s_1)$ | $\hat{h}_2(s_2)$ | $\hat{h}_2(s_3)$ | $\cdots$ |
| $\hat{h}_3$ | $\hat{h}_3(s_0)$ | $\hat{h}_3(s_1)$ | $\hat{h}_3(s_2)$ | $\hat{h}_3(s_3)$ | $\cdots$ |
| $\vdots$ | $\vdots$ | $\vdots$ | $\vdots$ | $\vdots$ | $\ddots$ |
| $f$ | $\Delta\big(\{\hat{h}_0(s_0)\}\big)$ | $\Delta\big(\{\hat{h}_j(s_1) \mid j \leq 1\}\big)$ | $\Delta\big(\{\hat{h}_j(s_2) \mid j \leq 2\}\big)$ | $\Delta\big(\{\hat{h}_j(s_3) \mid j \leq 3\}\big)$ | $\cdots$ |

## 3.2 Any Computable LLM will Hallucinate

In this section, we go one more step towards a direct answer to the fundamental question by considering LLMs as general total computable functions. This step can be achieved by using Theorem 1 and 2. Any *individual* computable LLM $h$ forms a set $\{h\}$, which only contains $h$ itself and is thus computably enumerable. Therefore, applying Theorem 1 on this set, we know that there exists a ground truth function $f$ such that all states of this particular $h$ will hallucinate. Furthermore, applying Theorem 2, we know that some $f$ can make hallucination happen for infinitely many inputs. This can be done for any total computable LLM, hence the following theorem similar to (Gold, 1967):

**Theorem 3.** For all computable LLMs $h$, there exists a computable ground truth function $f$ such that each of $h^{[j]}, j \in \mathbb{N}$ hallucinates w.r.t. $f$. Furthermore, there exists a computable ground truth function[3] $f'$ such that each of $h^{[j]}, j \in \mathbb{N}$ hallucinates on infinitely many inputs w.r.t. $f'$.

As a specific example, by adapting results from (Gold, 1967), we present a special case where $f$ represents a computable linear order, which is an abstraction of many real-world relations such as ranking of movies or songs, chronological ordering of events, alphabetical sorting of words, and so on.

**Theorem 4.** For all computable LLM $h$, there exists a computable ordering $<$ such that LLM $h$ hallucinates when answering question "$s_{2n+1} < s_{2n}$?" after being trained on training samples $\{(\ ``s_i < s_j?", f_<(s_i < s_j?)) \mid i, j < 2n\}$, where $n \in \mathbb{N}$, $f_<(s_i < s_j?) = $ "yes" if $s_i < s_j$ and "no" otherwise.

Appendix D provides proof, analysis, and empirical study for this theorem.

An obvious but important corollary of Theorem 3 is that it is impossible to use an LLM to eliminate its hallucination:

**Corollary 1.** All computable LLMs cannot prevent themselves from hallucinating.

If there exists an LLM $h'$ which can prevent itself from hallucination, then for any computable function $f$, there exists a state of $h'$ which is hallucination-free w.r.t. $f$, which contradicts Theorem 3. It explicitly suggests that methods relying on LLMs themselves to mitigate hallucination, such as prompt-based chain of thoughts (Wei et al., 2022b), cannot *eliminate* hallucination.

**Answer to the Fundamental Question** It has been shown that hallucination is inevitable for (1) LLMs in any computably enumerable set (Theorem 1 and 2), and (2) any total computable LLMs (Theorem 3). Since the formal world is a part of the real world, and real-world LLMs are a part of total computable LLMs, we conclude with a negative answer to the fundamental question: *hallucination of LLMs cannot be eliminated in the real world.* This is independent of implementation details such as (1) LLMs' architecture, (2) training procedure, (3) input prompts and questions, and so on, as long as they follow the definitions in Section 2 (which all real-world LLMs follow).

## 4 Discussion

### 4.1 Identifying Hallucination-Prone Problems

The idea behind all the theorems in Section 3 is that, for given (sets of) LLMs, we find computable ground truth function $f$ that all of the LLMs cannot learn, and therefore they will inevitably hallucinate therein. Therefore, one way to identify hallucination-prone problems is to identify what cannot be computed by the given set of LLMs. Table 3 provides an incomplete list of such problems. In each problem listed in the table, ground truth $f$ is expected to have the following behavior:

- Combinatorial List: $f$ lists all the strings with length $n$ using an alphabet of two characters. Computing $f$ takes $\Omega(2^n)$ time.

---

[3]In this theorem $f$ and $f'$ differs for different $h$, while $f$ in Theorem 1 and 2 is constructed for all LLMs in the set being considered.

Table 3: Identified hallucination-prone problems for different sets of LLMs. $O(n^k)$ and $O(2^n)$ indicate polynomial-time and exponential-time complexity, respectively.

| LLMs | Problem | Hallucination Guaranteed by | Extra Assumption |
|---|---|---|---|
| $O(n^k)$ time bounded LLMs (e.g., all existing LLMs) | Combinatorial List | Requiring $\Omega(2^n)$ solution | None |
| | Subset Sum | Being NP-complete | P $\neq$ NP |
| | Boolean Satisfiability (SAT) | Being NP-complete (Cook, 1971) | P $\neq$ NP |
| | Entailment of Propositional Logic | Being co-NP-complete (Arora & Barak, 2009) | P $\neq$ NP |
| $O(2^n)$ time bounded LLMs, $O(n^k)$ time bounded LLMs | Presburger Arithmetic (Presburger, 1929; Stansifer, 1984) | Requiring $\Omega(2^{2^{cn}})$ solution (Fischer & Rabin, 1974) | None |
| All Computable LLMs (incl. all LLMs above) | Learning all Computable Linear Orders | Theorem 4 | None |
| | Solving all Computable Problems | Theorem 3 | None |
| | Entailment of First-Order Logic | Being Undecidable (Turing, 1937) | None |

- Presburger arithmetic (Presburger, 1929; Stansifer, 1984): Presburger arithmetic is the first-order theory of natural numbers with addition and order $<$. Given a statement in the arithmetic, $f$ returns "yes" if the statement can be proved within the arithmetic and "no" otherwise. Computing $f$ takes $\Omega(2^{2^{cn}})$ time, for some $c > 0$ (Fischer & Rabin, 1974).

- Subset Sum: given a set of $n$ integers and a number $q$, $f$ returns "yes" when there is a subset that sums up to $q$ and "no" otherwise. This problem is NP-complete.

- Boolean Satisfiability (SAT) (Cook, 1971): given a formula of $n$ Boolean variables, $f$ returns "yes" if there exists an assignment on these variables which results in the formula to be true and "no" otherwise. This problem is NP-complete (Cook, 1971).

- Entailment of propositional logic: let $M(\psi)$ and $M(\phi)$ be the set of all the models of propositional logic formula $\psi$ and $\phi$ respectively. $f$ returns "yes" if $M(\psi) \subseteq M(\phi)$ and "no" otherwise. This problem is co-NP-complete (Arora & Barak, 2009).

- Entailment of first-order logic: let $M(\psi)$ and $M(\phi)$ be the set of all the models of first-order logic formula $\psi$ and $\phi$ respectively. $f$ returns "yes" if $M(\psi) \subseteq M(\phi)$ and "no" otherwise. This problem is undecidable (Turing, 1937).

As a result, real-world LLMs' answers about mathematical problems and logic reasoning should always be subject to proper scrutiny. The result is supported by a recent study (Hong et al., 2024). In Appendix C, we provide an empirical study to illustrate the combinatorial list problem. In Appendix D, we detail the problem of learning all computable linear orders and provide empirical results.

Furthermore, problems equivalent to those listed in Table 3 are also hallucination-prone. As a result, LLMs' answers about mathematical and logic reasoning should be subject to proper scrutiny.

## 4.2 Existing and Possible Hallucination Mitigators

We discuss efficacies and limitations of existing and possible hallucination mitigators as follows.

**Larger models, model ensembles, and more training data** LLMs are believed to show "emergent abilities" (Wei et al., 2022a) because of the significantly increased number of parameters and training samples. Therefore, it is natural to think that hallucination will disappear as LLMs grow larger. In the context of this work, increasing model parameters boosts the complexity of LLMs, enabling it to capture more complex ground truth functions. Increasing the size of $\mathcal{T}$ helps rule out invalid LLM candidates and helps with

convergence during training (Valiant, 1984). However, increasing parameters and data is futile if the ground truth function $f$ cannot be captured by LLMs at all as suggested in Section 3. For example, adding multi-head attention layers to a polynomial-time LLM will result in a larger polynomial-time LLM and will only reduce and possibly eliminate hallucination w.r.t. polynomial-time ground truth functions. It will not eliminate hallucination w.r.t. an exponential-time ground truth function, no matter how many layers or training data are added. Similarly, a model ensemble is essentially a single LLM. It is bounded by Theorem 3 and cannot eliminate hallucination.

**Prompting LLMs with Chain of Thoughts/Reflections/Verification**   This appoach belongs to the larger family of in-context learning, which provides example solutions or relevant knowledge about a target task in the prompt (Dhuliawala et al., 2023; Turpin et al., 2023; Wei et al., 2022b; Yao et al., 2023). Essentially, its intuition is that $f$ generates answer in a procedural manner. Complex problems often have multiple solutions with varying degrees of complexity. Prompting is effective in mitigating hallucination by guiding them towards solutions more preferred by humans, which are possibly the ones with lower complexities, and within LLMs' capabilities. For example, a recursive approach to compute Fibonacci sequence requires exponential time, whereas dynamic programming solves the same problem in linear time. However, it is unlikely that all ground-truth functions $f$ can be fully described by prompts. Therefore, this approach will only work for specific tasks. Moreover, as suggested by Corollary 1, it is impossible to eliminate hallucination by changing the prompts and hope LLM can automatically eliminate its hallucination.

**Guardrails and Fences**   The guardrails are principles that align LLMs' output with human values, ethics, and legal requirements. The fences is a list of critical tasks that should never be fully automated using LLMs. Both of them serve as safety constraints to prevent LLMs (and other AI models) from generating undesirable outcomes. Guardrails and fences can be formally programmed (Rebedea et al., 2023) to explicitly affect LLMs' behaviours. Therefore, it is more powerful than training samples defined in Definition 3 and is potentially a useful hallucination mitigator for the formal world and some real-world problems. However, its scalability in the real world remains an open problem.

**Knowledge-Enhanced LLMs**   This approach uses external knowledge (e.g., knowledge graphs and databases) and symbolic reasoning (e.g., logics) to aid LLMs both in training and inference. Popular chatbots driven by LLMs, such as ChatGPT, has started utilizing tools such as search engine, code intepreter, and calculators to solve complex problems beyond their innate capabilities (OpenAI, 2023). Similar to programmable guardrails and fences, it explicitly controls the LLM workflow by changing how information is recalled through retrieval from knowledge database (Chen et al., 2024; Lewis et al., 2020; Martino et al., 2023; Peng et al., 2023). In this way, LLMs receive extra information about the ground truth function $f$ other than via training samples. Therefore, Theorem 3 is inapplicable herein. This is potentially an effective mitigator of hallucination in the formal world. However, its scalability in real-world tasks is an open problem.

### 4.3   Practical Implications

We believe the discussions so far have the following implications for AI practitioners.

*All LLMs trained only with input-output pairs will hallucinate when used as general problem solvers.* This is guaranteed by Theorem 3. Problems beyond LLMs' computation capabilities are hallucination-prone. Note that a problem could be intellectually simple (for human) but computationally hard (for LLMs), and vice versa. So it is not immediately obvious whether a problem is (not) hallucination-prone for LLMs. Furthermore, computation complexity is only one of the many reasons for hallucination in the real world. For example, imperfect training data can lead to hallucination even in computationally easy tasks.

Another side note is that this implication only applies to "useful" LLMs, in the sense that the LLM must eventually give an answer for a question outside of its training data. Specifically, it can refuse to answer an arbitrary number of questions (e.g., by saying "I don't know") before it answers one. Whenever an LLM (in a state) answers a question outside of the training data it has seen, diagonalization technique in proofs for Theorem 1-2 can be applied to that answer. If an LLM never answers, then it will never hallucinate. On the other hand, as long as it answers for some unseen questions, it will hallucinate in some formal world.

*Without external aids like guardrails, fences, knowledge base, and human control, LLMs cannot be used automatically in any safety-critical decision-making.* This is a direct corollary of the conclusion above. External aids could help LLMs overcome limitations stated in Section 3 as it provides information beyond training samples of input-output pairs. Furthermore, humans' role is paramount because human values are arguably more complex than validity. Making safety-critical decisions usually requires rational and humane (e.g., understanding, empathy, and ethicality) judgements that are otherwise very difficult, if not impossible, to compute. Any errors due to hallucination in these cases, for example, making decisions about human life, are very likely to be unacceptable.

*Research and regulations on LLMs' safety boundaries is crucial and urgent in ensuring sustainable development of LLMs.* Real-world financial loss has been reported, where LLMs used for customer service provides incorrect information (BBC, 2024). Hallucination in automated sensing and actuation, such as in a robotics setting, might result in dangerous real-world outcomes. It is therefore important for theorists and practitioners to reach consensus on the ability boundaries for LLMs. There should also be appropriate regulations to prevent LLM uses outside of their ability boundaries. A preliminary attempt towards an "upper bound" of LLMs' capabilities is presented in Appendix E.

### 4.4 Limitations

While this work has answered a fundamental question about hallucination, we note its limitations. *First*, it does not account for hallucinations on problems within LLMs' computational capabilities. We believe the reasons behind hallucination is multifaceted; formal discussion is one facet that we feel is under-emphasized and under-explored. *Second*, it assumes deterministic ground truth function and thus provides little insights from the probabilistic perspectives. *Third*, in empirical study, existing LLMs are used without being further finetuned. Nevertheless, we believe these limitations are not critical to the question answered in the paper, but constitute open problems in Appendix H.

## 5 Related Works

This section provides a concise overview of pertinent studies in hallucination in LLMs. Recent surveys (Huang et al., 2023a; Ji et al., 2023; Luo et al., 2024; Pan et al., 2023; Wang et al., 2023a) provide extensive reviews of the field. We also discuss briefly the relation between this paper, PAC learnability, and online learnability in Appendix G.

### 5.1 Classification of Hallucination

A conventional classification of hallucination is the intrinsic-extrinsic dichotomy (Dziri et al., 2021; Huang et al., 2023b; Ji et al., 2023; Zhang et al., 2023). Intrinsic hallucination occurs when LLM outputs contradict with the provided input, such as prompts. On the other hand, extrinsic hallucination occurs when LLM outputs cannot be verified by the information in the input. Huang *et al.* (Huang et al., 2023a) extends this dichotomy by introducing faithfulness hallucination. Rawte *et al.* (Rawte et al., 2023) divided hallucination into "factual mirage" and "silver lining", denoting erroneous outputs based on factually correct or incorrect inputs.

### 5.2 Causes of Hallucination

Hallucination has been generally attributed to issues in data, training, and inference stages as identified in existing surveys (Huang et al., 2023a; Ji et al., 2023). Issues in the data include poor quality (Lee et al., 2022a), misinformation (Lin et al., 2022), bias (Narayanan Venkit et al., 2023; Paullada et al., 2021), and being outdated (Li et al., 2023a; Onoe et al., 2022) knowledge. Moreover, a fair portion of knowledge in the wild is long-tailed, making it challenging to recall during deployment (Kandpal et al., 2023; Mallen et al., 2023). During training, the model could suffer from architectural and strategic deficiencies which hinder proper learning such as exposure bias (Bengio et al., 2015) and diluted attention (Chiang & Cholak, 2022; Hahn, 2020; Liu et al., 2023) During inference, hallucination can also be caused by sampling randomness (Aksitov et al.,

2023; Dziri et al., 2021), insufficient context attention (Shi et al., 2023a), and softmax bottleneck (Chang & McCallum, 2022; Yang et al., 2018).

### 5.3 Mitigating Hallucination

The mitigation of hallucination involves tackling its underlying causes. For data-related issues, solutions include fact-focused datasets (Gao et al., 2020; Gunasekar et al., 2023) and developing automatic data cleaning techniques (Nie et al., 2019; Raunak et al., 2021; Shen et al., 2021). Retrieval augmentation, which uses relevant external documents to ground LLMs, can help reduce knowledge gap and reduce hallucination (Shuster et al., 2021; Zhao et al., 2023). Prompting techniques like Chain-of-Thought (Wei et al., 2022b) and Tree-of-Thought (Yao et al., 2023) have been applied to improve knowledge recall and reasoning (Wang et al., 2023b). To mitigate training-related hallucination, architectural improvements and training objectives like sharpening softmax functions (Liu et al., 2023) and factuality-enhanced training objectives (Lee et al., 2022b; Shi et al., 2023b) have been proposed. To overcome inference-related hallucination, new decoding methods are introduced to improve factuality or faithfulness of LLMs. Factual-nucleus sampling, proposed by Lee *et al.* (Lee et al., 2022b), aims to balance the diversity and the factuality in model outputs. Chain-of-Verification, introduced by Dhuliawala *et al.* (Dhuliawala et al., 2023), prompts the LLM to self-correct their mistakes during generation.

## 6 Conclusion

In this paper, we study the fundamental problem of eliminating hallucinations in LLMs. To do so, we define a formal world where hallucination in LLMs can be clearly defined and discussed. Specifically, hallucination is defined as inconsistencies between computable LLMs and a computable ground truth function. By utilizing results in learning theory, we show that hallucination is inevitable for computable LLMs if the ground truth function is any computable function. Since the formal world is a part of the real world, we further conclude that it is impossible to eliminate hallucination in the real world LLMs. We discuss the possible mechanisms and effectiveness of existing hallucination mitigators and discuss practical implications that our theoretical results have on the deployment of LLMs in the real world. We emphasize that since hallucination is inevitable, rigorous study of the safety of LLMs is critical and urgent.

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

# Appendix

## A   Notation and Terms

Table A1: Table of Notation and Terms.

| Functions | |
|---|---|
| $f$ | A ground truth function. |
| $h$ | An LLM. |
| $h^{[n]}$ | A state of LLM $h$, where it is trained on the $0^{th}, 1^{st}, \ldots, (n-1)^{th}$ training samples. |
| $\{h_0, h_1, \ldots\}$ | A computably enumerable set of LLMs. |
| $\hat{h}_k$ | The $k^{th}$ entry in an enumeration of all the states of LLMs in the set $\{h_0, h_1, \ldots\}$, where for $k = (i+n)(i+n+1)/2 + n$, $\hat{h}_k$ is the same LLM as $h_i^{[n]}$, $i, n \in \mathbb{N}$. |
| **Data** | |
| $\mathcal{A}$ | The finite alphabet of tokens. |
| $w_{0:n-1}$ | Alternative notation of string $w_0 w_1 \ldots w_{n-1}$, where $w_i \in \mathcal{A}, i \in \mathbb{N}$. |
| $\mathcal{S}$ | The set of all finite-length strings made from tokens in $\mathcal{A}$. |
| $s_i$ | The $i^{th}$ string of an one-to-one computable enumeration $(s_0, s_1, \ldots,)$ of $\mathcal{S}$. |
| $\mathcal{G}_f$ | The set $\{(s_i, f(s_i)) \mid i \in \mathbb{N}\}$ of all input-output pairs of function $f$ on $\mathcal{S}$. |
| $\mathcal{T}_n$ | The set of training samples $\{(s_i, f(s_i)) \mid i = 0, 1, \ldots, n-1\}$. |
| **Other Notations** | |
| $p(w_{0:n-1})$ | Probability of string $w_{0:n-1}$. |
| $\Delta(s)$ | A string different from $s$. |
| $\Delta(\{s_0, s_1, \ldots\})$ | A string different from all the strings in the set $\{s_0, s_1, \ldots\}$. |
| $L(m, \mathcal{A})$ | A task that requests all the strings of length $m$ made of tokens in alphabet $\mathcal{A}$. |
| $\omega(m)$ | A task that asks if a statement about a linear order is true, after providing examples of $m$ elements in that linear order. |
| $R(m, n)$ | A task that asks for the $n^{th}$ character of the input string with a length of $m$ characters. |
| **Terms** | |
| Total Computable Function on $\mathcal{S}$ | A function that produces an output in finite steps for each input in $\mathcal{S}$. |

## B   Implementation of Large Language Models

LLMs are essentially probabilistic models of strings made of tokens, where the likelihood of each token is conditioned on all the tokens that come before it in the string. Formally, the likelihood of a string $w_0 w_1 \ldots w_{n-1}$ is

$$p(w_{0:n-1}) = p(w_0) \prod_{t=1}^{n-1} p(w_t | w_{0:t-1}), \ w_i \in \mathcal{A}, n \in \mathbb{N}. \tag{B1}$$

The most common usage of LLM is to complete a partial string $w_0 w_1 \ldots w_{q-1}$, or a "prompt", into a complete string $w_0 w_1 \ldots w_{n-1}$, by maximizing the likelihood of the latter:

$$p(w_{0:n-1}) \propto p(w_{q:n-1} | w_{0:q-1}) = \prod_{t=q}^{n-1} p(w_t | w_{0:t-1}), \ w_i \in \mathcal{A}, q, n \in \mathbb{N}, q < n. \tag{B2}$$

As the equation suggests, such completion occurs iteratively, one token at a time. In practice, the iteration stops after a fixed number of iterations, or when a special stopping token is generated.

State-of-the-art LLMs (Touvron et al., 2023a;b; Brown et al., 2020; OpenAI, 2023) are generally implemented using the Transformer architecture, which is a multi-layer feedforward network equipped with attention mechanism (Vaswani et al., 2017). The "large" in LLM describes both the scale of the models, which is usually billions of parameters, and the size of training corpora, which is usually trillions of tokens (Touvron et al., 2023b; OpenAI, 2023). An LLM is trained with the objective that the output string $w_{1:n-1}$ resembles the real-world corpora and is coherent and reasonable. Common training procedures include (1) unsupervised pretraining where LLMs are trained to complete strings from general corpus auto-regressively, (2) supervised finetuning where the pretrained LLMs are finetuned on training samples for specific tasks, and (3) reinforcement learning where human feedback is put as training signals for societal value alignment.

## C  Empirical Study: Can LLMs List Them All?

Consider a task intellectually simple for human beings: list all strings of a fixed length using a fixed alphabet. This task is listed in Table 3, which stated that polynomial-time LLMs will hallucinate on this task. In this section, we empirically validate this statement.

**The Task**   An LLM is required to list all the strings with length $m$ using an alphabet $\mathcal{A}$. We denote this task as $L(m, \mathcal{A})$. For example, a solution to $L(2, \{\texttt{a}, \texttt{b}\})$ is "bb, ba, ab, bb". For each $L(m, \mathcal{A})$, we run the LLM three times using three random seeds. The LLM is deemed successful in solving $L(m, \mathcal{A})$ if, in any of the three runs, the LLM's output contains all and only strings of length $m$ using alphabet $\mathcal{A}$. Note that duplicated entries are allowed in the output, for example, "aa, ab, ba, bb, ab" is a valid solution to $L(2, \{\texttt{a}, \texttt{b}\})$.

**Models**   In the experiment we study two LLM families, namely Llama 2 (Touvron et al., 2023b), Llama 3 (Meta Platforms, Inc.), and Generative Pretrained Transformers (GPT) (Brown et al., 2020; OpenAI, 2023). Models in both families have context windows of at least 4096 tokens. For Llama 2, we use its 70-billion-parameter version `llama2-70b-chat-hf` publicly accessible on HuggingFace (HuggingFace, a). For Llama 3, we use its 70-billion-parameter four-bit quantized version `llama3-70B-Instruct-bnb-4bit` (HuggingFace, c). For GPT, we use the 175-billion-parameter `gpt-3.5-turbo-16k` for GPT-3.5, `gpt-4-0613` for the GPT-4 family with more parameters (GPT-4's parameter number is not disclosed in the report (OpenAI, 2023)), and the latest GPT-4 variant `gpt-4-turbo-2024-04-09`.

Table C2: Evaluation results of LLMs on $L(m, \{\texttt{a}, \texttt{b}\})$ and $L(m, \{\texttt{a}, \texttt{b}, \texttt{c}\})$ tasks. A check mark ✓ indicates that the LLM successfully solved the problem and a cross mark ✗ indicates the opposite. The exact parameter number for GPT-4 is not disclosed. While `gpt-4-turbo-2024-04-09` was successful in $L(5, \{\texttt{a}, \texttt{b}, \texttt{c}\})$, it failed an equivalent task $L(5, \{\texttt{x}, \texttt{y}, \texttt{z}\})$, hence the asterisked ✓*.

| LLM | # Param | # Context (tokens) | $L(m, \{\texttt{a}, \texttt{b}\})$ | | | | | | | $L(m, \{\texttt{a}, \texttt{b}, \texttt{c}\})$ | | | | |
|---|---|---|---|---|---|---|---|---|---|---|---|---|---|---|
| | | $m \rightarrow$ | 1 | 2 | 3 | 4 | 5 | 6 | 7 | 1 | 2 | 3 | 4 | 5 |
| `llama2-70B-chat-hf` | $7.00 \times 10^{10}$ | 4,096 | ✓ | ✓ | ✗ | ✗ | ✗ | ✗ | ✗ | ✓ | ✗ | ✗ | ✗ | ✗ |
| `llama3-70B-Instruct-bnb-4bit` | $7.00 \times 10^{10}$ | 8,000 | ✓ | ✓ | ✓ | ✗ | ✗ | ✗ | ✗ | ✓ | ✓ | ✓ | ✗ | ✗ |
| `gpt-3.5-turbo-16k` | $1.75 \times 10^{11}$ | 16,385 | ✓ | ✓ | ✓ | ✓ | ✓ | ✗ | ✗ | ✓ | ✓ | ✓ | ✗ | ✗ |
| `gpt-4-0613` | $\geq 1.75 \times 10^{11}$ | 8,192 | ✓ | ✓ | ✓ | ✓ | ✓ | ✓ | ✗ | ✓ | ✓ | ✓ | ✓ | ✗ |
| `gpt-4-turbo-2024-04-09` | $\geq 1.75 \times 10^{11}$ | 128,000 | ✓ | ✓ | ✓ | ✓ | ✓ | ✓ | ✗ | ✓ | ✓ | ✓ | ✓ | ✓* |

**Prompt**   We first prompt LLMs to complete a dialogue made of questions and answers. The following is used as the base prompt for all the empirical studies in this paper. It is adapted from (HuggingFace, b) with modifications made to prompt the LLMs to answer directly instead of providing high-level descriptions or code snippets.

**Base Prompt**

```
You are a helpful, respectful and honest assistant. Always answer as helpfully as possible, while being safe. Your
answers should not include any harmful, unethical, racist, sexist, toxic, dangerous, or illegal content. Please ensure
that your responses are socially unbiased and positive in nature.

If you don't know the answer to a question, please don't share false information. However, if you know the answer, you
should always share it in every detail and as requested. Always answer directly. Do not respond with a script or any
approximation.
```

Then, we describe the task to the LLM, for example, $L(2, \{\mathtt{a}, \mathtt{b}\})$ is described as:

**Task Description for $L(2, \{\mathtt{a}, \mathtt{b}\})$**

```
List ALL the strings with a length of two characters and only contains characters "a" and "b". Do not miss a single
string.
```

**Result**  We feed the base prompt and the task descriptions for $L(m, \{\mathtt{a}, \mathtt{b}\})$ and $L(m, \{\mathtt{a}, \mathtt{b}, \mathtt{c}\})$ with $m \in \mathbb{N}$ to the LLMs. The evaluation result is shown in Table C2. Consistent with claims in Table 3, all LLMs failed eventually as $m$ grows. Surprisingly, LLMs fail for tasks with short answer, compared to the length of their context window. For example, the answer of $L(7, \{\mathtt{a}, \mathtt{b}\})$ is a list of $2^7 = 128$ strings, each with 7 characters. This results in an 896-character-long answer, which is much shorter than the context window of tested LLMs. Furthermore, the number of parameters and context size does not significantly impact LLMs' performance on this task: they are equally poor.

## D  LLM Cannot Learn All Computable Linear Orders

As a supplement to Theorem 3, this section presents a concrete example of a computable function representing a linear order between strings. A linear order is a binary relation between strings, where for each pair of distinct strings $(s_i, s_j)$, either $s_i < s_j$ or $s_j < s_i$. The ordering relation is transitive and irreflexive. This relation is an abstraction of many real-world relations such as ranking of movies or songs, chronological ordering of events, alphabetical sorting of words, and so on. We further provide an empirical study following the theoretical results.

We consider only linear orders which are isomorphic to $\mathbb{N} = 0, 1, 2, \ldots$ or $\mathbb{Z} = \ldots, -3, -2, -1, 0, 1, 2, \ldots$. Since the order is assumed to be computable, there is a ground truth function $f_<$ which determines if $s_i < s_j$ for any $s_i$ and $s_j$. We define $f_<(s_i < s_j?) = \mathtt{"yes"}$ if $s_i < s_j$ and $\mathtt{"no"}$ otherwise.

For ease of presentation, in this section, we abuse the notation of $h^{[n]}$ to denote the state of LLM $h$ trained on $\{(\text{ "}s_i < s_j?\text{"}, f_<(s_i < s_j?)) \mid i, j < 2n\}$.

### D.1  Theoretical Results

We present the following theorem that LLM cannot learn all computable linear orders, by adapting results from (Gold, 1967). The theorem has been presented as Theorem 4 in Section 3, we restate it below for the ease of presentation:

**Theorem.** For all computable LLM $h$, there exists a computable ordering $<$ such that LLM $h$ hallucinates when answering question "$s_{2n+1} < s_{2n}?$" after being trained on training samples $\{(\text{ "}s_i < s_j?\text{"}, f_<(s_i < s_j?)) \mid i, j < 2n\}$, where $n \in \mathbb{N}$, $f_<(s_i < s_j?) = \mathtt{"yes"}$ if $s_i < s_j$ and $\mathtt{"no"}$ otherwise.

*Proof.* We now construct a computable ordering $<$ where $h^{[n]}$ hallucinates for all $n$. Such ordering is constructed in stages, where in stage $n$, we define $<$ on all pairs $s_i, s_j$ with $i, j < 2n + 2$.

Let $\mathcal{T}_n$ be the training samples $\{(\text{ "}s_i < s_j?\text{"}, f_<(s_i < s_j?)) \mid i, j < 2n\}$. Increasing $n$ is equivalent to providing more training samples to the model. Let $h^{[n]}$ denote the LLM that has been trained on $\mathcal{T}_n$.

1. Start from stage 0.

2. By induction, by stage $n - 1$, relation $<$ has been defined between $s_i, s_j$ with $i, j < 2n$. So in stage $n$, we only need to define the ordering between $s_{2n}$, $s_{2n+1}$, and $s_i, i < 2n$, as below:

(a) If $h^{[n]}(s_{2n+1} < s_{2n}?) = $ `"yes"`, then

Define $s_i < s_{2n} < s_{2n+1}$, for $i < 2n$.

As a result, $f_<(s_{2n+1} < s_{2n}?) = $ `"no"` $\neq h^{[n]}(s_{2n+1} < s_{2n}?)$.

(b) If $h^{[n]}(s_{2n+1} < s_{2n}?) = $ `"no"`, then

Define $s_i < s_{2n+1} < s_{2n}$, for $i < 2n$.

As a result, $f_<(s_{2n+1} < s_{2n}?) = $ `"yes"` $\neq h^{[n]}(s_{2n+1} < s_{2n}?)$.

3. Go to stage $n + 1$.

Then:

1. $<$ is computable because: (1) it is defined for all pairs of $(s_i, s_j)$ and is defined using computation results from a computable LLM $h$, and (2) it is defined for all pairs $(s_i, s_j)$ with $i, j \leq 2n + 1$ either at or before stage $n$.

2. For all $n$, $h^{[n]}(s_{2n+1} < s_{2n}?) \neq f_<(s_{2n+1} < s_{2n}?)$ because we explicitly defined $<$ to contradict $h^{[n]}$ in step 2a and 2b for all $n$.

$\square$

As a result, an LLM will inevitably hallucinate when determining linear orders between strings in the controlled world. This result holds for any $n$, where the examples given to $h$ is the ordering between any pair if $(s_i, s_j)$, with $i, j < 2n$.

Furthermore, it turns out that LLMs will make mistakes infinitely often when answering if a linear order is isomorphic to $\mathbb{N}$ or $\mathbb{Z}$, as shown in (Bazhenov et al., 2020) for general learning algorithms:

**Theorem D1.** For all computable LLM $h$, there exists a computable ordering $<$ such that $h$ makes errors when answering question "is ordering $<$ isomorphic to $\mathbb{N}$ or $\mathbb{Z}$ ?" after being trained on $\{(\text{"}s_i < s_j?\text{"}, f_<(s_i < s_j?)) \mid i, j < 2n\}$, for infinitely many $n$, where $n \in \mathbb{N}$, $f_<(s_i < s_j?) = $ `"yes"` if $s_i < s_j$ and `"no"` otherwise.

*Proof.* We now construct a computable ordering $<$ where for infinitely many $n$, $h^{[n]}$ gives the wrong answer. Such ordering is constructed in stages, where in stage $n$, we define $<$ on all pairs $s_i, s_j$ with $i, j < 2n + 2$.

1. Start from stage 0.

2. By induction, in stage $n - 1$, relation $<$ has been defined between $s_i, s_j$ with $i, j < 2n$. So in stage $n$, we only need to define the ordering between $s_{2n}$, $s_{2n+1}$, and $s_i, i < 2n$, as below:

(a) If $h^{[n]}(\text{"}\mathbb{N} \text{ or } \mathbb{Z}?\text{"}) = \mathbb{N}$, then

Define $s_{2n+1} < s_i < s_{2n}$, for $i < 2n$.

(b) If $h^{[n]}(\text{"}\mathbb{N} \text{ or } \mathbb{Z}?\text{"}) = \mathbb{Z}$, then

Define $s_i < s_{2n} < s_{2n+1}$, for $i < 2n$.

3. Go to stage $n + 1$.

Then:

1. Relation $<$ is computable: (1) it is defined for all pairs of $(s_i, s_j)$ and is defined using computation results from a computable LLM $h$, and (2) it is defined for all pairs $(s_i, s_j)$ with $i, j \leq 2n + 1$ either at or before stage $n$.

2. The ordering given by $<$ is either isomorphic to $\mathbb{N}$ or $\mathbb{Z}$ because in every stage, we always add element above all elements in the finite order constructed in the previous stages, and sometimes below all these elements.

3. For infinite number of $n$, $h^{[n]}$ will answer the question wrongly, because

    (a) If $h^{[n]}$ gives the answer $\mathbb{N}$ for infinitely many $n$, then step 2a is taken infinitely often, and the linear order is isomorphic to $\mathbb{Z}$.

    (b) If $h^{[n]}$ gives the answer $\mathbb{Z}$ for almost all $n$, then step 2b is taken almost always, and the linear order is isomorphic to $\mathbb{N}$.

Thus, $h$ gives wrong answers infinitely often. $\qquad\square$

Theorem 4 and D1 show that computable LLMs will hallucinate on questions about both local and global properties of linear ordering $<$, no matter how many training samples are provided.

## D.2   Empirical Study

In this section, we explore the capabilities of LLMs to discern linear order relations between binary integers. To prevent the LLMs from reciting the corpus and really "reason" using the provided information, we use an intentionally perplexing scheme. Specifically, symbol "$" is used to represent the relation "less than", or "<" as generally written in corpus. Furthermore, character "b" is used to represent "0", and "a" for "1". A linear order "0<1" in natural language thus translates to "b$a" in our abstract representation.

**The Task**   An LLM is provided with examples describing orders between binary numbers from zero (i.e., binary number $0_2$ represented by b) up to $m$ (e.g., binary number $10000_2$ represented by abbbb). It is then asked if a statement "$x\$y$" is true, where $x$ and $y$ are strings representing binary numbers. This task is denoted $\omega(m)$.

There are two cases in $\omega(m)$: (1) both $x$ and $y$ are present in the samples, but their relation is not, and (2) at least one of $x$ and $y$ is not in the samples. We denote these two cases as $\omega_1(m)$ and $\omega_2(m)$, respectively. For each case, we randomly generate five pairs of strings for test. For each pair $(s_i, s_j)$, we generate two statements "$s_i\$s_j$" and "$s_j\$s_i$". This results in ten test statements for each case.

We run LLM on these two cases using three random seeds. The LLM is deemed successful in solving $\omega_*(m)$ if, in any of the three runs, it correctly determine if each of the test statements is true. Furthermore, for $\omega_2(m)$, the LLM is also deemed successful if its answer is "unknown" for all the statements.

**Models**   We use the same LLMs and configurations as in Appendix C.

**Prompt**   The prompt begins with the formulation of the strings and the introduction of a partial order relation "$" among them, as follows:

---
**Definition of relation**

```
There is a relation "$" between strings made of characters "a" and "b".

Given such strings x, y, and z, the relation has the following properties:
(a) if x$y is true, then y$x is false,
(b) if both x$y and y$z are true, then x$z is true, and
(c) x$x is always false, for any x.
```

Following the properties, a list of samples satisfying the relation is provided.

---
**Examples for $\Omega(m)$**

```
Below are examples of strings that satisfy the relation "$".
Note that many other strings also satisfy the relation.
b$a
a$ab
ab$aa
/* omitted for conciseness, the examples cover binary integers from 0 to m */
...
(omitted)
```

We finally ask the question using the following:

Table D3: Evaluation results of LLMs on $\omega_*(m)$ tasks. A check mark ✔ indicates that the LLM successfully solved the problem and a cross mark ✗ indicates the opposite. The exact parameter number for GPT-4 is not revealed.

| LLM | # Param | # Context (tokens) | $\omega(1000_2)$ | | $\omega(10000_2)$ | |
|---|---|---|---|---|---|---|
| | | Cases → | $\omega_1(1000_2)$ | $\omega_2(1000_2)$ | $\omega_1(10000_2)$ | $\omega_2(10000_2)$ |
| `llama2-70B-chat-hf` | $7.00 \times 10^{10}$ | 4,096 | ✗ | ✗ | ✗ | ✗ |
| `llama3-70B-Instruct-bnb-4bit` | $7.00 \times 10^{10}$ | 8,000 | ✗ | ✗ | ✗ | ✗ |
| `gpt-3.5-turbo-16k` | $1.75 \times 10^{11}$ | 16,385 | ✗ | ✗ | ✗ | ✗ |
| `gpt-4-0613` | $\geq 1.75 \times 10^{11}$ | 8,192 | ✗ | ✗ | ✗ | ✗ |
| `gpt-4-turbo-2024-04-09` | $\geq 1.75 \times 10^{11}$ | 128,000 | ✗ | ✗ | ✗ | ✗ |

---

**Task Description**

```
Given the above information, determine if {string 1}${string 2} is true.

If it is true, your answer must be "true".
If it is false, your answer must be "false".
If you do not know if it is true or false, you answer must be "unknown".
```

---

**Result**   The results are shown in Table D3. Shockingly but expectedly, all LLMs failed in this task. For $\omega_1$ tasks, examination of LLMs' responses reveal that they are unable to use the transitive rule to deduce relations. For $\omega_2$ tasks, LLMs tend to give inconsistent answers for "$x\$y$" and "$y\$x$", for example, giving "unknown" for one but "true" for the other. This suggests a deficiency in their reasoning abilities.

## E  Identifying Limits on LLMs' Capabilities

One may naturally think the other way round: what are the functions on which an LLM can possibly be hallucination-free? We give the following theorem as an "upper bound" of the capability of LLMs. We show that the set of functions on which LLM is hallucination-free is contained in a computably enumerable set of total computable functions, by adapting results and proofs from (Bārzdiņš & Freivald, 1972) and (Gold, 1967):

> **Theorem E2.** Let set $\mathcal{F}$ denote the set of all the ground truth functions w.r.t. which a computable LLM $h$ can be hallucination-free after being trained on some finite number of training samples. Then $\mathcal{F}$ is contained in a computably enumerable set of total computable functions.

*Proof.* LLMs are trained on training samples. Therefore, to study what LLMs can reproduce, we need to first study all the possible training sets. We enumerate training sets of one sample, two samples, three samples, and so on. For example, training sets with one sample can be enumerated as

$$\left(\{(s_0, y_0)\}, \{(s_1, y_1)\}, \dots\right).$$

Training sets with two samples can be enumerated following the Cantor pairing function which maps a pair of natural number to a natural number:

$$\left(\{(s_0, y_0), (s_1, y_1)\}, \{(s_0, y_0), (s_2, y_2)\}, \{(s_1, y_1), (s_2, y_2)\}, \{(s_0, y_0), (s_3, y_3)\}, \dots\right).$$

We can then continue to list training sets of $n = 3, 4, \dots$ samples to get all the possible training sets.

Let $\mathcal{T}_n^i$ denote the $i^{th}$ training set that contains $n$ samples. We can list samples in it as $\mathcal{T}_n^i = \{(s_{i,0}, y_{i,0}), (s_{i,1}, y_{i,1}), \dots, (s_{i,n-1}, y_{i,n-1})\}$, where $n > 0$ and $y_{i,*}$ is the expected output for $s_{i,*}$ under some ground truth function. For example, $\mathcal{T}_1^0 = \{(s_0, y_0)\}$ and $\mathcal{T}_2^1 = \{(s_0, y_0), (s_2, y_2)\}$ according to our enumeration above.

Let $h_n^i$ be an LLM trained on $\mathcal{T}_n^i$. For each $\mathcal{T}_n^i$, define a function $f_n^i$ as follows:

$$\forall s \in \mathcal{S}, f_n^i(s) = \begin{cases} y_{i,j} & \text{if } s = s_{i,j} \text{ for some } j < n, \\ h_n^i(s) & \text{otherwise.} \end{cases} \tag{E3}$$

Let $\mathcal{F}' = \{f_n^i \mid i, n \in \mathbb{N}\}$. We show (a) $\mathcal{F} \subseteq \mathcal{F}'$ followed by (b) $\mathcal{F}'$ is computably enumerable.

(a) First, we show that $\mathcal{F} \subseteq \mathcal{F}'$.

- Consider any $f \in \mathcal{F}$. According to the definition of $\mathcal{F}$, there exists $i, n \in \mathbb{N}$ such that $h_n^i$ is hallucination-free w.r.t. $f$, so $\forall s \in \mathcal{S}, h_n^i(s) = f(s)$.

- $h_n^i$ is trained on $\mathcal{T}_n^i$, so $h_n^i(s) = y_{i,j}$ if $s = s_{i,j}$ for some $j < n$. Therefore, by the definition of $f_n^i$, $\forall s \in \mathcal{S}, f_n^i(s) = h_n^i(s)$.

- Therefore, $\forall s \in \mathcal{S}, f_n^i(s) = f(s)$. This means that such ground truth $f$ is $f_n^i$.

Repeating this reasoning for all $i, n \in \mathbb{N}$, we find that any $f \in \mathcal{F}$ must be some $f_n^i \in \mathcal{F}'$. Therefore, $\mathcal{F} \subseteq \mathcal{F}'$.

(b) Then, we show that $\mathcal{F}'$ is a computably enumerable set of total computable functions. First, every $f_n^i$ is total computable because LLM $h_n^i$ is total computable. Second, set $\{\mathcal{T}_n^i \mid i, n \in \mathbb{N}\}$ is computably enumerable and we have defined each $f_n^i$ for a $\mathcal{T}_n^i$, so there is an one-to-one mapping between each $\mathcal{T}_n^i$ and $f_n^i$, which means $\mathcal{F}'$ is computably enumerable. As a result, $\mathcal{F}'$ is a computably enumerable set of total computable functions.

Therefore, $\mathcal{F}$ is contained in a computably enumerable set of total computable functions. $\qquad \square$

Theorem E2 offers another view to our negative answer to the fundamental question: there are many real-world problems whose answers are not computable at all, let alone being contained in a computably enumerable set of total computable functions, and therefore cannot be solved by LLMs. For example, the problem of deciding if any program will halt is not computable.

In defence for LLMs, we present the following theorem that some ground truth functions can be learned by some LLMs, also by adapting results from (Bārzdiņš & Freivald, 1972) and (Gold, 1967):

> **Theorem E3.** For all computably enumerable sets $\mathcal{C}$ of total computable functions, there exist an LLM $h$ such that for all computable functions $f \in \mathcal{C}$, there exists a training set of up to $k$ samples $\{s_0, s_1, \ldots, s_k\}$, such that $h^{[k]}$ is hallucination free on $f$.

Theorem E3 implies that there are some real-world problems where LLMs can be trained to be hallucination-free, as long as their ground truth functions are contained in a computably enumerable set of total computable functions.

For example, the task of returning the $n^{th}$ character of input string of length $m \in \mathbb{N}$ is learnable by some (and not all) LLMs. Denote this task as $R(m, n)$. We consider the following cases $\{R(m, n) \mid m \in \{16, 64, 256, 1024\}, n \in \{1, 2, 5\}\}$. For every $n$, we provide LLMs discussed in Appendix C with five examples showing the $n^{th}$ character of some randomly generated 16-character strings. The LLMs are then tested on a different set of five randomly generated strings. Each LLM is tested three times using different random seeds. An LLM is considered successful in a case if, in any of the three runs, it correctly returns the $n^{th}$ character for all five test strings in that case. As shown in Table E4, all LLMs performed well in $R(*, 1)$. However, Llama 2 fails $R(*, 2)$ and all of the LLMs fails $R(m, 5)$ for $m \geq 256$. The result is surprising and implies that (1) training corpora have not covered enough training samples for counting and indexing and/or (2) state-of-the-art LLMs have not achieved the theoretical limits of their abilities. This calls for further research into training techniques and architecture design for LLMs.

Table E4: Evaluation result for the task of $R(m, n)$, i.e., returning the $n^{th}$ character of the input string of $m$ characters. A check mark ✓ indicates that the LLM successfully solved the problem and a cross mark ✗ indicates the opposite. The exact parameter number for GPT-4 is not revealed.

| LLM | # Context (tokens) | $R(m, 1)$ | | | | $R(m, 2)$ | | | | $R(m, 5)$ | | | |
|---|---|---|---|---|---|---|---|---|---|---|---|---|---|
| | $m \rightarrow$ | 16 | 64 | 256 | 1024 | 16 | 64 | 256 | 1024 | 16 | 64 | 256 | 1024 |
| `llama2-70B-chat-hf` | 4,096 | ✓ | ✓ | ✓ | ✓ | ✗ | ✗ | ✗ | ✗ | ✗ | ✗ | ✗ | ✗ |
| `llama3-70B-Instruct-bnb-4bit` | 8,000 | ✓ | ✓ | ✓ | ✓ | ✓ | ✓ | ✓ | ✓ | ✓ | ✓ | ✗ | ✗ |
| `gpt-3.5-turbo-16k` | 16,385 | ✓ | ✓ | ✓ | ✓ | ✓ | ✓ | ✓ | ✓ | ✗ | ✗ | ✗ | ✗ |
| `gpt-4-0613` | 8,192 | ✓ | ✓ | ✓ | ✓ | ✓ | ✓ | ✓ | ✓ | ✗ | ✗ | ✗ | ✗ |
| `gpt-4-turbo-2024-04-09` | 128,000 | ✓ | ✓ | ✓ | ✓ | ✓ | ✓ | ✓ | ✓ | ✓ | ✓ | ✗ | ✗ |

# F In Defence of LLMs and Hallucination

Although we have shown that LLMs will inevitably hallucinate, this does not undermine their tremendous value in enhancing productivity. Moreover, hallucination itself should not be viewed entirely negatively. In this section, we would like to defend the usage of LLMs and their hallucinatory inclinations.

First, the decision to use an LLM is fundamentally a trade-off between precision and efficiency, which is largely determined by their application. LLMs excel in processing and structuring information at scales and speeds unachievable by humans, facilitating rapid decision-making and idea generation. In scenarios where speed and volume of information processing are overwhelming, the occasional inaccuracies of LLMs are acceptable compromises. Conversely, in situations where precision is critical, the outputs of LLMs can (and must) be verified and supplemented with human supervision. It is notable that while LLMs cannot learn *all* computable ground truth functions $f$, it can learn *some* classes of $f$ (see Appendix E) and can be useful therein. The key is not to view LLMs as infallible sources of truth but as powerful assistive tools for information retrieval, analysis, summarisation, and presentation.

Moreover, it is crucial to recognize that LLMs are continuously evolving. Such evolution includes advancements in model architecture, data processing techniques, training methodologies, and error correction strategies. Over time, we anticipate that the nature of hallucination will be better understood by human researchers and users. While it is impossible to completely eliminate hallucination, we are optimistic that its severity may be controlled and reduced for many applications.

Finally, hallucination is not completely detrimental. In art, literature, and design, the unintentional, unpredictable, or even nonsensical outputs from LLMs could inspire human creators. Such deviation from facts can lead to unique perspectives that a strictly factual and precise system might never generate. In this sense, the hallucinatory aspect of LLMs should be regarded positively as a source of inspiration, innovation, and creativity.

# G Comparison with PAC and Online Learnability

This section briefly discusses the difference between the analysis in this paper and learnability in PAC and online learning theories. Specifically, we explain why PAC and online unlearnability do not answer Question 1.

## G.1 PAC Unlearnability does not answer Question 1

Our formal world is defined by computable functions. The set of all computable functions is not PAC learnable. This translates to the following statement:

**Statement 1.** There exists no $O\left(\text{Polynomial}\left(\frac{1}{\epsilon}, \frac{1}{\delta}\right)\right)$-time algorithm that can find an LLM that has $\epsilon$ or lower hallucination rate in all formal worlds with probability $(1 - \delta)$, for all distributions over input strings, and for all $0 < \epsilon, \delta < \frac{1}{2}$.

Theorem 1-2 in this paper translate to the following statement:

**Statement 2.** There is no computably enumerable sets of LLMs that has no hallucination in all formal worlds.

Statement 1 and Statement 2 are different. Statement 1 describes practical learnability within certain time complexity and error rate, while Statement 2 describes learnability regardless of those assumptions. Therefore, Statement 2 is more relevant to Question 1.

### G.2 Online Unlearnability does not answer Question 1

In this paper, hallucination is not directly linked to unbounded mistakes, and thus differs from the conventional definition of being not online learnable. An LLM can be hallucination-free while making an unbounded (but finite) number of mistakes.

We show that there exists a class $\mathcal{F}$ that has infinite Littlestone dimension (Littlestone, 1987), but there exists an LLM, when trained using Procedure 1, will not hallucinate on it. Consider $\mathcal{F} : \{f : f(s) = 0 \text{ for almost every s}, s \in \mathcal{S}\}$. In other words, $\forall f \in \mathcal{F}$, $f(s) \neq 0$ for an arbitrary but finite number of $s \in \mathcal{S}$. $\mathcal{F}$ has an infinite Littlestone dimension, thus resulting in unbounded mistakes in the online learning setting. However, an LLM as used in this paper can learn this class by predicting the zero-extension of the input seen so far in the training. There exists an state of this LLM which would not make any error beyond the point when all non-zero points for $f$ has been provided in the training. In this case, this LLM does not hallucinate.

Therefore, conventional notions of online unlearnability does not answer Question 1.

## H Open Problems

Finally, we list some challenging and fundamental open problems stemming from our discussion:

- What real-world problems can be provably addressed by LLMs without hallucination? The problem can be further divided: (1) what is the theoretical computational capability of an LLM, (2) what are the complexities of real-world problems, and (3) how can LLMs achieve their theoretical capabilities so that hallucination can be eliminated in limited realms?

- How can hallucination be detected and corrected for real-world problems using external knowledge sources and reasoning tools? Can LLMs find the appropriate external tools for solving real-world problems? Should we train an LLM that does everything or an LLM that can use proper tools for specific problems?

- How to quantify the risk of hallucination for real-world problems? Quantifying hallucination risk is an important step to identity on which problems can LLMs be safely deployed, even with inevitable hallucination. Probabilistic methods could be helpful in bounding such risk.

## I Notes on Empirical Study

All the empirical studies presented in this paper were conducted between December 2023 and May 2024. Inference of Llama 2 was conducted using four NVIDIA A100 GPUs. Inference of four-bit quantized version of Llama 3 was conducted using one NVIDIA A100 GPU.

