# OpenReview forum: "Hallucination is Inevitable: An Innate Limitation of Large Language Models"
_TMLR — Rejected by TMLR_

### Review · Reviewer_JoHq · 2024-11-17

**Summary Of Contributions:**

This paper tackles the problem of hallucination in Large Language Models (LLMs).
It presents a formal framework defining LLMs and hallucination within a "world" of computable functions, where hallucination is defined as inconsistency between an LLM's output and a ground truth computable function.
The authors claim to prove that hallucination is inevitable for any computable LLM, regardless of architecture, training data, or prompting techniques.
The authors then discuss the implications of this result for existing and potential hallucination mitigation strategies, arguing that relying solely on LLMs for mitigation, such as through chain-of-thought prompting, is insufficient.
Finally, empirical evidence is presented that demonstrates the failure of several LLMs on seemingly simple tasks that require listing all strings of a given length and discerning linear orders.

**Audience:**

No

**Broader Impact Concerns:**

Overall comment: I am not a mathematician, so I may be missing some crucial details that makes this paper a solid contribution. However, I question it's relevance and I would like for the community to avoid providing "peer-reviewed" evidence for click-baity titles such as "Scientists prove that LLMs will always hallucinate". I think at the very least the title should be changed to correspond to what is shown in the paper, e.g. "one may **construct** _hypothetical_ functions and their inputs such that the LLMs will not be able to determine correct/expected result on them, if you use a the LLM's outptu to construct these functions". In particular, I would argue that if the LLM would answer "I don't know", this would not be a "hallucination" as the authors define it.

**Claims And Evidence:**

No

**Requested Changes:**

My main concerns regarding this paper are the assumptions on which the presented framework relies upon and definitions of notions such as "hallucinations". Here I list them all.

Critical points to be addressed for recommendation for acceptance:
1. The authors write "since the formal world is a part of the real world which is much more complicated, hallucinations are also inevitable for real world LLMs.". is this really a problem on real-world tasks? humans hallucinate as well too, no? Therefore, can it be that LLMs could be "allowed" to have some degree of hallucinations in corner cases yet still be deemed as useful?
2. You write: "The trained LLM h[i] can be expected to: (1) answer f (s) =?, or (2) given “(s,” complete it with “f (s))”, or (3) given “is it true that f (s) = t ?” answer “yes” or “no”. For ease of presentation and without loss of generality, we assume case (1), where LLM uses h[i](s) to answer the question." I don't see what these 3 cases are brining. It is also unclear to me what they represent. How do they differ? Why is it relevant to present them here? Isn't it enough to see the LLM as a black-box function that, given an input s, produces an output f(s)?
3. "It is noteworthy that LLMs trained by Procedure 1 are far more powerful and flexible than their counterparts in the real world.". Why is this true? I don't see what makes this procedure so special? Please provide evidence for this claim.
4. "for example, we could define ∆ ( s k ) = s k+1, which returns the next string of s k in S". I think you should not use \delta for this. It is too confusing, I was expecting some delta difference, choose better some other function name. Also in the table you write \delta(h(s)) but here it is \delta(s)? You are also specifying it as a function of h(s)? But that does not correspond to reality. Reality is that this function is simply a function of s. Why would an LLM not learn it in that case? This is in a way adversarial to the LLM, if you "look" at its outputs once it is trained, and then _construct_ a GT function with different outputs... It could be that I'm missing some BG knowledge, but I really don't understand how this construction "proves" that LLMs _will_ hallucinate. Therefore, I leave others to judge this merit.
5. "One might argue that Theorem 1 only shows that all states of all the LLMs in any computably enumerable set will hallucinate on one but not all inputs." - why is this the case? Didn't you write "Repeating this reasoning for all s i ∈S, we see that all states of all the LLMs in the given set will hallucinate w.r.t."?
6. "Theorem 2: ..there exists a computable ground truth function f , such that all h[j ] i ,i,j ∈ N, hallucinate on infinitely many inputs." - even if this is the case, it's questionable how many of these infinitely many inputs are of _real world_ significance.
Example: when generating an image, you can generate _infinitely_ many meaningless images (acc. to humans) if you simply sample each pixel from Gaussian distribution. You will get something that resembles a TV "noise" when it is out of signal.
It is extremely unlikely that you will ever get images that humans find "interesting" and that correspond to real-world. Instead, real world images seem to live on a low-dimensional manifold. Therefore, even if an LLM would hallucinate on infinitely many examples, as long as it is correct on the low-dimensional manifold of real-world images, it is "perfect" for real-world use cases.
7. You define "hallucinations" as when the LLM output differs from the ground-truth/correct output. However, if an LLM would answer "I don't know" when it cannot compute an answer, I wouldn't call that a "hallucination". The same is true for humans. Your arguments look to me more that an LLMs can't be guaranteed to be able to correctly solve an NP-complete problem.
8. "suggests that methods relying on LLMs themselves to mitigate hallucination, such as prompt-based chain of thoughts (Wei et al., 2022b), cannot eliminate hallucination." - even if this sentence is true (though you use the careful wording of "suggest") and the CoT can't help correcting LLMs output, what about tool use?
9. Your combinatorial list task evaluation "f lists all the strings with length n using an alphabet of two characters. Computing f takes Ω(2n ) time." - the LLM could write a python code that would be able to execute this (provided enough time)? Humans would also not be able to do this task, yet we would not say that "humans provably hallucinate"...


Minor points:
1. "For example, in the survey paper (Ji et al., 2023), the authors attribute hallucination in natural language generation to heuristic data collection, innate divergence, imperfect representation learning, erroneous decoding, exposure bias, and parametric knowledge bias" - every point here should have a citation associated with it.
2. Figure 2: Remark: maybe put "==" in the formula in the (d) drawing to better indicate comparison.
3. Empirical study is presented as a contribution, but all findings are in the Appendix. Ideally, it should be integrated in the main paper sections.
4. "learning such as exposure bias (Bengio et al., 2015) and diluted attention (Chiang & Cholak, 2022; Hahn, 2020; Liu et al., 2023)" - missing a dot at the end.

**Strengths And Weaknesses:**

Strengths:
1. the paper studies an important problem
2. the paper is mostly well written

Weaknesses:
1. I'm not sure if the main assumptions on which the theory in this paper is built upon are valid (see comments below).

---

> ### Author Response · Authors · 2024-12-10
> **Author Response (1/3)**
>
> Thank you very much for your detailed and thought-provoking comments.
> We realize that our paper can be confusing and your comments will certainly help improve the exposition and will make the paper clearer.
> Please find below for our response to your specific questions and concerns.
> We look forward to further engagement with you.
>
> > Is this really a problem on real-world tasks? humans hallucinate as well too, no? Therefore, can it be that LLMs could be "allowed" to have some degree of hallucinations in corner cases yet still be deemed as useful?
>
> The paper does not claim that:
>
> (1) given a real-world problem, LLM must hallucinate therein, or
>
> (2) humans do not hallucinate, or
>
> (3) LLMs are not allowed to hallucinate, or
>
> (4) LLMs are useless because they inevitably hallucinate.
>
> What the paper claims is: for any computable LLM, we can find a ground truth function representing a computable problem on which this LLM must hallucinate. Since any computable problem is a part of the world, LLMs will inevitably hallucinate in the real world.
>
> This conclusion aligns with our definition of hallucination and proof presented in the paper.
> Furthermore, we have also listed real-world tasks where LLMs with reasonably constrained complexities must hallucinate.
>
> In spite of hallucinations, we strongly believe that LLMs are extremely useful and they definitely can be used in the real world.
> However, we point out some limitations and therefore suggest that LLMs be used with some care.
> The paper has elaborated on these issues in Section 4 and Section F.
>
> > You write: "The trained LLM h[i] can be expected to: ..." I don't see what these 3 cases are bringing. It is also unclear to me what they represent. How do they differ? Why is it relevant to present them here? Isn't it enough to see the LLM as a black-box function that, given an input s, produces an output f(s)?
>
> The paper used $h$ to describe an LLM, which is viewed as a black-box function; the only contraint on $h$ is that it must be computable.
>
> The text quoted by the reviewer describes the possible behaviors of LLM $h$ after being trained by Procedure 1.
> Case (1) says LLMs can be used to answer questions, where for every question $s$, the correct answer is determined by function $f$ and is unique.
> Case (2) says LLMs can be used to complete a partial input prompt, where $f(s)$ is a possible completion of $s$.
> If for every input prompt, the correct completion is unique, then case (1) and case (2) are equivalent.
> Case (3) says LLM can be simply only answering "yes" or "no" for questions.
> If time limit for LLM is unbounded, then case (3) is equivalent to case (1).
>
> As a result, case (1) is what we choose for this paper because: (a) we have assumed unique ground truth and (b) we put no time constraint on LLMs in the theorems.
>
>
> > "It is noteworthy that LLMs trained by Procedure 1 are far more powerful and flexible than their counterparts in the real world.". Why is this true? I don't see what makes this procedure so special? Please provide evidence for this claim.
>
> Practically, Procedure 1 is less constrained than any existing training procedure.
> As a result, LLMs trained using this procedure is more powerful than real-world ones.
> For example:
> (1) it allows LLMs to be trained on arbitrarily large but finite number of training samples before they are tested, and
> (2) it does not assume a specific optimizer, which means that one can assume, for example, the optimizer to be the ideal one that it trains LLMs to perform perfectly on the training dataset without introducing overfitting.
>
> Furthermore, in our definition 4 of hallucination, an LLM is considered hallucination-free if any of its states fully reproduces the ground truth function.
> This is different from real-world cases where LLMs are trained once and then evaluated.
>
> > "for example, we could define ∆ ( s k ) = s k+1, which returns the next string of s k in S". I think you should not use \delta for this. It is too confusing, I was expecting some delta difference, choose better some other function name. Also in the table you write \delta(h(s)) but here it is \delta(s)? You are also specifying it as a function of h(s)? But that does not correspond to reality. Reality is that this function is simply a function of s. Why would an LLM not learn it in that case?
>
> Function $\Delta$ was defined to return a different string from its input string.
> It could be renamed to $f_\Delta$ if that causes confusion.
> The output of $h$ given an input string $s$, or $h(s)$, is also a string, so it is a proper input to function $\Delta$.
>
> That $f$ is not learnable by LLMs is proved by diagonalization; we have provided detailed proof for the main theorems in the paper.

---

> > ### Author Response · Authors · 2024-12-10
> > **Author Response (2/3)**
> >
> > > This is in a way adversarial to the LLM, if you "look" at its outputs once it is trained, and then construct a GT function with different outputs…
> >
> > The ground truth function is defined just based on the computably enumerable list of LLMs (and their input/output behaviours), and we do not need to know which trained LLM is used.
> > The construction of $f$ is done before an LLM is trained.
> > It does not require knowledge about which LLM state is obtained at the end of a particular training session.
> > For more detailed response to this question, we would like to invite you to read our common response to all reviewers posted above.
> >
> > > "Theorem 2: ..there exists a computable ground truth function f , such that all h[j ] i ,i,j ∈ N, hallucinate on infinitely many inputs." - even if this is the case, it's questionable how many of these infinitely many inputs are of real world significance. Example: when generating an image, you can generate infinitely many meaningless images... Therefore, even if an LLM would hallucinate on infinitely many examples, as long as it is correct on the low-dimensional manifold of real-world images, it is "perfect" for real-world use cases.
> >
> > Thank you for pointing this out.
> > What you are saying is saying that LLMs could be hallucination-free in some specific families or instances of real-world problems, which we agree.
> > In fact, this is exactly what makes the LLMs so useful in the real world.
> >
> > On the other hand, what the proof of Theorem 2 shows is that for every LLM, there exists computable problems in the real world such that this LLM will hallucinate on an infinite number of inputs on this computable function.
> >
> > Unfortunately, we cannot know whether some of these hallucinatory inputs correspond to real-world situtations, which is very important to safety-critical applications of LLMs.
> >
> > > You define "hallucinations" as when the LLM output differs from the ground-truth/correct output. However, if an LLM would answer "I don't know" when it cannot compute an answer, I wouldn't call that a "hallucination". The same is true for humans. Your arguments look to me more that an LLMs can't be guaranteed to be able to correctly solve an NP-complete problem.
> >
> > An LLM could refuse to answer a question by saying "I don't know".
> > However, if an LLM answers questions only finitely often, then it almost always refuses to answer questions and is useless.
> > If the LLM answers questions infinitely often, then the proof in theorem 2 works on all its non-"I don't know" outputs.
> > We discussed this case in Section 4.3 at the bottom of page 9.
> >
> > > "suggests that methods relying on LLMs themselves to mitigate hallucination, such as prompt-based chain of thoughts (Wei et al., 2022b), cannot eliminate hallucination." - even if this sentence is true (though you use the careful wording of "suggest") and the CoT can't help correcting LLMs output, what about tool use?
> >
> > Thank you for pointing this out.
> > We actually recommend using external tools with LLMs.
> > We discussed tool use in Section 4.2 at the paragraph starting with "Knowledge-Enhanced LLMs".
> > We quote it below:
> >
> > _Popular chatbots driven by LLMs, such as ChatGPT, has started utilizing tools such as search engine, code intepreter, and calculators to solve complex problems beyond their innate capabilities (OpenAI, 2023). Similar to programmable guardrails and fences, it explicitly controls the LLM workflow by changing how information is recalled through retrieval from knowledge database (Chen et al., 2024; Lewis et al., 2020; Martino et al., 2023; Peng et al., 2023). In this way, LLMs receive extra information about the ground truth function $f$ other than via training samples. Therefore, Theorem 3 is inapplicable herein. This is potentially an effective mitigator of hallucination in the formal world. However, its scalability in real-world tasks is an open problem._
> >
> > > Your combinatorial list task evaluation "f lists all the strings with length n using an alphabet of two characters. Computing f takes Ω(2n ) time." - the LLM could write a python code that would be able to execute this (provided enough time)?
> >
> > The statement given by the reviewer is true if the LLM and the Python interpreter is given enough time to run.
> > On the other hand, in Table 3 we propose combinatorial list task as a problem where $O(n^k)$ (polynomial-time) LLMs will inevitably hallucinate.
> > If the Python interpreter is only given polynomial time, it will still fail for this problem.
> >
> > > Humans would also not be able to do this task, yet we would not say that "humans provably hallucinate"...
> >
> > The aim of the paper is just to say that all computable LLMs will hallucinate by the definition of this paper.
> > Our scope is limited to computable LLMs, and not humans.

---

> ### Author Response · Authors · 2024-12-10
> **Author Response (3/3)**
>
> > Overall comment: I am not a mathematician, so I may be missing some crucial details that makes this paper a solid contribution. However, I question it's relevance. I would like for the community to avoid providing "peer-reviewed" evidence for click-baity titles such as "Scientists prove that LLMs will always hallucinate".
>
> > I think at the very least the title should be changed to correspond to what is shown in the paper, e.g. "one may construct hypothetical functions and their inputs such that the LLMs will not be able to determine correct/expected result on them, if you use a the LLM's outptu to construct these functions".
>
> We completely agree that the scientific community should reject click-baity and ungrounded statements about anything, including LLMs.
>
> The construction of $f$ does not require $f$ seeing the output of a specific LLM after training.
> It is constructed based on an enumeration of a set of LLMs and their input-output behaviours.
> Please refer to our common response posted above.
>
> In our view, the title is appropriate, as the paper clearly defines what is a formal world, what is hallucination (as errors made by LLMs) in a formal world, and proved why for any LLM there exists a formal world where it will inevitably hallucinate.
> Since any formal world is a subset of the real world, the title holds in general.
>
> Our paper makes rigorous provable statements about LLM hallucination, and we stand by our title.

---

### Review · Reviewer_HSHP · 2024-11-18

**Summary Of Contributions:**

This paper intends to address the fundamental problem of hallucination in large language models (LLMs), defined as the generation of factually incorrect or nonsensical outputs. It introduces a formal framework to rigorously define and analyze hallucination, establishing that hallucination is unavoidable for all computable LLMs, regardless of architecture or training procedure. Using results from learning theory, the authors prove that no LLM can perfectly replicate every computable function, leading to inevitable hallucinations. The core claim is "LLMs will Hallucinate on Infinitely Many Questions". Empirical validation and practical implications on LLM deployment are discussed, along with evaluations of current mitigation techniques.

**Audience:**

Yes

**Claims And Evidence:**

No

**Requested Changes:**

1. On the top of page 3, you said, "All real-world LLMs have some properties, for example, they all complete their computation in polynomial time". However, in Figure 1, you draw P-property LLM as a subset of LLM. Aren't they contradictory?
2. In definition 2, you said, "f(s) is the only correct output of input string s". However, there are always multiple correct outputs for an input string. The output string "101" and "It's 101" are all correct answers for the example input “What is the sum of binary numbers 10 and 11?”. Both "Joe Biden" and "Joe Robinette Biden Jr." are correct answers to the question "Who is the 46th president in the US?". The definition of the ground truth is not accurate for LLMs and should be updated.
3. How would the theoretical results change if the ground truth function were probabilistic or context-dependent? For example, when we ask, "What is the result of flipping a fair coin?", the answer could be "0" or "1". Either might be correct with a 50% chance.
4. Above Table 2, what is the empty square at the right end?
5. Is there any example of "O(2^n) time bounded LLMs" or "O(n^k) time bounded LLMs"?

**Strengths And Weaknesses:**

Strength:
1. The paper introduces a formal world framework to define hallucination, bridging a gap in theoretical understanding.
2. The use of learning theory and diagonalization techniques strongly supports the idea that hallucination is inevitable for LLMs. The concept and its proof process are intriguing.
3. The results are applicable across most computable LLMs, independent of specific implementation details like architecture or training datasets.
4. It highlights critical implications for the safe deployment of LLMs, emphasizing the necessity of external safeguards in safety-critical applications.

Weaknesses:
1. The focus on a formal world may oversimplify real-world nuances.

Although there is only one weakness, it is a fundamental problem in the paper and is pretty serious. Details are in "Requested Changes".
Moreover, I select "No" in "Claims And Evidence" primarily because of the weakness. The main claim is not fully supported by the proof in the paper.

---

> ### Author Response · Authors · 2024-12-10
>
> Thank you for your very constructive comments.
> Please see our response to your questions below.
> We look forward to further engagement with you.
>
> > On the top of page 3, you said, "All real-world LLMs have some properties, for example, they all complete their computation in polynomial time". However, in Figure 1, you draw P-property LLM as a subset of LLM. Aren't they contradictory?
>
> They are not contradictory.
> An LLM in this paper is a computable function.
> It can be a real-world resource-constrained LLM, or a hypothetical one not bound by real-world constraints.
> Therefore, the set of all LLMs is a superset of all real-world LLMs.
> This ensures that our conclusion is applicable to either today's LLMs or those in the future.
>
> > There are always multiple correct outputs for an input string... The definition of the ground truth is not accurate for LLMs and should be updated.
>
> > How would the theoretical results change if the ground truth function were probabilistic or context-dependent? For example, when we ask, "What is the result of flipping a fair coin?", the answer could be "0" or "1". Either might be correct with a 50% chance.
>
> A formal world in this paper is a part of the real world but not the real world itself.
> A ground truth function of a formal world only has one correct output given an input, and it does not need to cover everything in the real world.
> Similarly, the probabilistic or context-dependent setting is not covered by the ground truth function, but it does not change the conclusion of this paper.
>
> The rationale of the paper is:
> (1) it defines formal worlds which are linked to computable problems, (2) any computable problem is a part of the real world, (3) LLMs will inevitably hallucinate in some computable problem, and therefore (4) LLMs will inevitably hallucinate in the real world.
>
> > Above Table 2, what is the empty square at the right end?
>
> This is the Q.E.D. symbol that marks the end of a proof.
> The symbol is defined by the `amsthm` package for the `proof` environment.
> Please refer to Section 5.2 of its documentation [1].
>
> > Is there any example of "O(2^n) time bounded LLMs" or "O(n^k) time bounded LLMs"?
>
> An LLM that list all strings from length 1 to length $k$, using an alphabet of $\\{0,1\\}$ is $O(2^n)$ time bounded.
> All transformer-based LLMs are $O(n^2)$ bounded, as a result, they are also $O(2^n)$ and $O(n^k)$ bounded for $k \geq 2$.
>
>
> **Reference**
>
> [1] _Using the amsthm Package_, https://www.ams.org/arc/tex/amscls/amsthdoc.pdf#page=11.77

---

### Review · Reviewer_z1tm · 2024-12-05

**Summary Of Contributions:**

The paper establishes that hallucination is an inherent limitation of large language models (LLMs). Using a formal framework, the authors demonstrate that LLMs inevitably produce inconsistencies when compared to a computable ground truth function. The theoretical analysis steps from specific to general, essentially proving that any computable LLM will hallucinate on  infinitely many inputs. The paper concludes by discussing hallucination-prone tasks and potential implications.

**Audience:**

Yes

**Claims And Evidence:**

Yes

**Requested Changes:**

I look forward to engaging with the authors to gain further clarity on the theoretical results presented in the paper.

**Strengths And Weaknesses:**

**Strengths**

* The paper addresses a fundamental and important question regarding the inevitability of hallucination in large language models (LLMs).
* The presentation is clear overall, with most concepts explained effectively.
* The discussion in Section 4 is thorough and illustrative, covering relevant works and practical implications.

**Weaknesses**
* My biggest concern is regarding the reasoning of the theoretical part. Specifically, it is unclear why the ground truth function is defined in terms of the predictions of an LLM. This raises questions about how the LLM can be trained under such assumptions. Why does the trained model seem to precede the ground truth, rather than the reverse?
* Why is Definition 4 constructed for the LLM $h$ as a whole, rather than for certain stages $h^[i]$?
* Minor point: $f(s_0)$ is not defined until Section 2.2, which could lead to confusion in earlier parts of the paper.

---

> ### Author Response · Authors · 2024-12-10
>
> Thank you for your insightful and positive comments.
> Please find our response to your questions and concerns.
> We look forward to further engagement with you.
>
> > Why does the trained model seem to precede the ground truth, rather than the reverse?
>
> The trained LLMs do not precede the definition of the ground truth function.
> We have explained this in detail in our common response to all reviewers posted above.
>
> > It is unclear why the ground truth function is defined in terms of the predictions of an LLM. This raises questions about how the LLM can be trained under such assumptions.
>
> The ground truth function is defined before an LLM is trained.
> It is one of the computable functions based on the given enumerable set of LLMs.
> The ground truth function depends on the enumerated set of LLMs, and thus their input/output behaviour.
> However, as explained above, the definition of the ground truth function does not need to know exactly which LLM is obtained after training.
>
> > Why is Definition 4 constructed for the LLM $h$ as a whole, rather than for certain stages $h^{[i]}$?
>
> Definition 4 is constructed for LLM $h$ as a whole  because we need to consider hallucination for the LLM $h$ when it is trained on all possible amounts of training data, each corresponding to one state of the LLM $h$.
> By definition 4, an LLM is hallucination-free as long as it has one state that accurately reproduces that ground truth function.
>
> In other words, if given $f$, there exists natural number $i$ such that $\\{(s_0,f(s_0)),(s_1,f(s_1)),\ldots,(s_i, f(s_i))\\}$ trains an LLM $h$ to a state $h^{[i]}$ that accurately reproduces $f$, then we say that LLM $h$ does not hallucinate on $f$.
> The proof for Theorems 1 and 2 construct $f$ such that there exists no such number $i$ for all LLMs in the given computably enumerable set.
>
> > Minor point: $f(s_0)$ is not defined until Section 2.2, which could lead to confusion in earlier parts of the paper.
>
> Thank you very much for pointing this out.
> We realize that $f(s_0)$ should have been introduced earlier when training samples were first mentioned (i.e., in Section 2.1).
> We will incorporate this point in the next version of the paper.

---

### Author Response · Authors · 2024-12-10
**Common Response to all Reviewers**

We thank all the reviewers for their time and effort in providing detailed feedback on our paper.
We have posted our response to the specific threads and look forward to further engagement with the reviewers.
Below we clarify a common question asked by the reviewers.

**Does a trained LLM precede the ground truth function? Or, are we looking at the output of a trained LLM before defining what $f$ should output?**

No, the trained LLM does not precede the ground truth function.

Defining the ground truth function in theorem 1 and 2 only requires knowledge about the computably enumerable set of LLMs that we are considering (and their input-output behaviours), but not the specific state of the LLM after training.
The ground truth function only needs the enumeration of the set of LLMs to determine what to output.

For example, a computable enumerable set of LLMs would be "the set of LLMs made of a natural number of fully-connected layers, where the number of neurons in each layer is a natural number and all of the weights representable by finite precision floating point numbers", which includes all LLMs realizable up till now.
The definition of $f$ only needs to enumerate all LLMs in this set (basically, to assign an index to each LLM in this set by the number of layers, the number of neurons, and the bits of each weight, which means we know their input-output behaviours) and determine its output based on the output of the LLM with that index.
No matter which LLM is obtained at the end of training, it must be one of the LLMs in that set, thus must correspond to one of the indices.
The definition of $f$, by diagonalization, ensures that no matter which index the trained LLM corresponds to, it must hallucinate on $f$.

It is obvious and less interesting that if someone gives us a trained LLM then we can find a ground truth function $f$ on which the LLM must hallucinate.
However, what is interesting and important is that if someone describes a computably enumerable set of LLMs to us then we can find _one_ ground truth function on which _all_ the LLMs in that set must hallucinate, regardless of the amount of training samples provided by $\\{(s_0,f(s_0)),(s_1,f(s_1)),\ldots\\}$.

---

### Decision · Action_Editor_kLYc · 2025-02-03

**Recommendation:** Reject

**Comment:**

Given the unanimous agreement amongst the reviewers, and my own concerns about the clarity of the work, in particular with how it defines hallucination. I recommend rejection.

**Audience:**

Many people are concerned about hallucination in large language models, so in that sense the paper discussed a topic of broad interest.

**Claims And Evidence:**

There is unanimous agreement between the reviewers that the paper fails to back up its claims with clear and convincing evidence.

According to the authors, the primary claim of the paper is:

> What the paper claims is: for any computable LLM, we can find a ground truth function representing a computable problem on which this LLM must hallucinate. Since any computable problem is a part of the world, LLMs will inevitably hallucinate in the real world.

I think I agree with the reviewers the the concept of "hallucination" the paper considers is divorced enough from what people in the field mean when they talk about hallucination to make the claim, even if true in the narrow sense in which it is defined in the paper, potentially misleading in the way it is framed and presented.